# Hydrometeorology and landscapes control sediment and dissolved organic carbon mobility across a diverse and changing glacier-sourced river basin

Craig A. Emmerton<sup>1,2</sup>\*, John F. Orwin<sup>3</sup>, Cristina Buendia<sup>1α</sup>, Michael Christensen<sup>4</sup>, Jennifer A. Graydon<sup>1</sup>, Brian Jackson<sup>5</sup>, Elynne Murray<sup>5</sup>, Stephanie Neufeld<sup>4</sup>, Brandi W. Newton<sup>3</sup>, Ryan Ozipko<sup>5</sup>, Rick Pickering<sup>5</sup>, Nadine Taube<sup>3</sup>, Chris Ware<sup>5</sup>

#### Author info and affiliations:

- \*corresponding author; <a href="mailto:craig.emmerton@gov.ab.ca">craig.emmerton@gov.ab.ca</a>
- <sup>a</sup>now at: Catalan Water Agency/Agència Catalana de l'Aigua

- 1. Government of Alberta, Alberta Environment and Protected Areas, 9th Floor 9888 Jasper Avenue, Edmonton, Alberta, Canada T5J 5C6
- 2. Department of Biological Sciences, University of Alberta, CW405 Biological Sciences Building, Edmonton, Alberta, Canada T6G 2E9
- 3. Government of Alberta, Alberta Environment and Protected Areas, 3535 Research Rd NW Calgary, Alberta, Canada T2L 2K8
- 4. Epcor Water Services, 2000-10423 101st St. NW, Edmonton, Alberta, Canada T5H 0E8
- 5. Government of Alberta, Alberta Environment and Protected Areas, 4816 89th St., Edmonton, Alberta, Canada T6E 5K1

#### Short summary:

Rivers are valuable indicators of climate change when extensively monitored. We used an integrated monitoring program within a changing river basin to understand how sediment and dissolved organic carbon change across differing landscapes and runoff conditions. We show that delivery of sediment and dissolved organic carbon changes widely between years and rivers draining pristine or impacted catchments. This work demonstrates challenges facing river water users under a changing climate.

#### **Abstract**

Northern landscapes are enduring ongoing impacts of anthropogenic land use and climate change. Rivers are valuable indicators of this change reflected by the timing and amount of water and terrestrial material they mobilize. Assessing the influence of a changing climate on diverse river systems is best achieved using multi-annual monitoring and replication of effort across varied tributary catchment conditions. We used this approach to monitor concentrations, catchment yields, and export of total suspended sediments (TSS) and dissolved organic carbon (DOC) of a large, diverse, glacial river network (North Saskatchewan River; NSR) in western Canada during years of extensive weather fluctuations. Though concentrations of TSS and DOC increased eastward through the NSR basin from Rocky Mountain cordillera to agriculturalized plains, catchment yields were statistically highest from cordillera regions, reflecting an eastward rain shadow. Wet conditions across the basin resulted in variable but statistically higher TSS and DOC yields compared to drought conditions. During wetter years, we observed disordered, threshold-type, erosive mobilization of TSS through the basin whereas DOC increased more predictably with runoff. Variability of yields and export was substantial both within and between pristine and impacted catchments, and within the NSR mainstem illustrating the complexity of river sediment and DOC transport at the network scale. Consequently, in a warming and wetting climate, we expect TSS and DOC transport to intensify with sediment transport being more difficult to predict compared to DOC, which has implications for aquatic ecosystems and >1.5M people who depend on the NSR for drinking water.

#### 1. Introduction

Northern latitudes have reported considerable human-driven climate and landscape changes post industrial revolution. Air temperatures have increased there at twice the global rate since the 1980s (Gulev et al., 2021) and extreme heat events have become more frequent (Simpkins, 2017; La Sorte et al., 2021; Vogel et al., 2019). Severe meteorological events in the form of high intensity rainfalls and droughts are also becoming more common (Walsh et al., 2020). Consequently, northern regions are losing glacial ice and snow cover (Marzeion et al., 2014; Mudryk et al., 2020), experiencing acute high and low water events (Pomeroy et al., 2016; Yin et al., 2018), and enduring more frequent and devastating wildfires (Hanes et al., 2018). The effect of these climate-driven shifts on regional hydrology and subsequent interaction with human-impacted landscapes (Brandt et al., 2013; Hirsh-Pearson et al., 2022) has resulted in marked changes in the transport of terrestrial material from landscapes to fluvial systems (Lu et al., 2013).

Changes in export of material by rivers is a key metric to identify and assess landscape response(s) to changes in climate and anthropogenic activities (Allan, 2004). This understanding is driven by the strong coupling between hydrometeorological conditions and the ability of rivers to mobilize landscape material, as modified by its availability. Notably, total suspended sediment (TSS) and dissolved organic carbon (DOC) are ubiquitous, landscape-derived materials found in rivers that are mobilized during runoff events and are sourced from erosion and flushing of upland soils, wetlands, riparian areas, and channel stores. Sediment transport typically increases in rivers as runoff and erosion regimes become more energetic in warming and wetting climates, particularly where sediment availability is increased as a result of anthropogenic activities or changes in vegetation cover (Whitehead et al., 2009). Flushing of agricultural soils and urban areas is known to drive nutrient, DOC, and contaminant delivery downstream (Giri and Qiu, 2016; St-Hilaire et al., 2016). At the other extreme, heat and drought can induce runoff shortages and low river flows that produce sediment- and organic-poor waters dominated by groundwater inputs (Mosley, 2015). Similarly, runoff through intact landscapes and wetlands can result in efficient soil and biological water storage and promotion of physical and biogeochemical removal of nutrients, DOC, and contaminants (Knox et al., 2008; Neary et al., 2009). However, hydrometeorological conditions and the mobilization of landscape material can vary significantly across temporal and spatial scales making it challenging to use rivers as indicators of change, especially where monitoring data may be sparse or unrepresentative (Orwin et al., 2010; Shi et al., 2017).

Long-term environmental monitoring programs are key tools for investigating the impacts of multiple stressors on fluvial systems (Lovett et al., 2007) and can address deficits common in shorter-term studies. For example, understanding climate impacts on rivers requires consistent monitoring to account for sometimes considerable interand intra-annual variability in hydrometeorological conditions. Different landscapes have varied and often complex influences on river water quality (Maloney and Weller, 2011), thus monitoring programs with sampling and replication of diverse conditions is beneficial (Lintern et al., 2018a). Further, using a nested river network monitoring design provides a means to differentiate contributions of landscape material (via tributary catchments) from in-river sources (i.e., banks, channels) of major rivers downstream (Chalov and Ivanov, 2023). Monitoring programs that include these features are rare but present an opportunity to better understand how climate variability and change across diverse landscapes can impact major rivers and water users; a key gap in our understanding of river water quality (van Vliet et al., 2023).

Here, we concurrently assess how widely varying hydrometeorological and landscape conditions of a diverse and changing northern river basin impact the downstream delivery of key water quality, aquatic ecosystem health, and drinking water treatment parameters (i.e., TSS, DOC). The North Saskatchewan River (NSR) in western Canada is sourced from the glaciated Canadian Rocky Mountains and transects mountain bedrock, forest, agricultural, and urbanized landscapes. In addition to ongoing changes in summer flows and sediment yields experienced by Rocky Mountain rivers (Goode et al., 2012; Rood et al., 2008), the NSR basin has recently (i.e., 2019-2022) endured hydrometeorological extremes, ranging from high runoff years to historic heat and drought conditions. These wide-ranging conditions may become more common in the future and thus provide a window to better understand the response of northern rivers to climate extremes. We used water quality and quantity

data from a nested monitoring program for this study. The WaterSHED (Water: Saskatchewan Headwaters Edmonton Downstream) Monitoring Program is a collaborative tributary catchment monitoring network across the NSR basin initiated in 2019 by a drinking water treatment utility, local governments, and a non-government organization (Emmerton et al., 2023). The program was designed to balance monitoring across different catchment landscape conditions within the NSR basin. In addition, long-term water quality and flow monitoring data were used from stations on the downstream NSR mainstem, operated by the governments of Alberta and Canada. We had two goals of this study: 1. to distinguish the impacts of interannual hydrometeorological variability from changes in landscape conditions on catchment yields of TSS and DOC: and 2. to determine the relative roles of catchment and in-channel contributions of a major river (i.e., NSR) on interannual export of TSS and DOC from the basin. We hypothesized that TSS (i.e., particulates) and DOC (i.e., organic material) yields would show similar interannual responses to changes in hydrometeorology across the network, but differences between landscape types. We also hypothesized that the degree of influence of in-channel sources and sinks of TSS and DOC relative to catchment inputs would change between wetter and drier years.

#### 2. Methods

#### 2.1 Setting

The upper NSR basin in Alberta, Canada is a large, diverse, continental basin draining ~40,000 km² of headwater landscapes and high prairie (Figure 1). The NSR originates at the Saskatchewan Glacier in Banff National Park in the Rockies. As the river travels east, it passes through cordillera, forested foothills, mixed forests, agriculturalized plains, and an urban centre (Edmonton; population: 1.5M; Statistics Canada, 2023) that sources drinking water from the NSR. Regulated mean annual discharge of the NSR at Edmonton is ~140 to 295 m³ s⁻¹. Surficial geology across the basin progresses from exposed bedrock in the Rockies to heterogeneous glacial tills in foothills and plains. This till has been shaped by climate, topography, and biogeochemistry since deglaciation ca. 14,000 years ago, resulting in sparse and thin luvisols in cordillera and foothills regions to thicker, organic- and nutrient-rich

Figure 1 The upper North Saskatchewan River (NSR) basin in Alberta, Canada showing tributary catchment and NSR mainstem water quality and flow stations. Catchment structural units (CSU; see Methods section 2.2.1) showing general landscape cover and land use across the basin is shown. Landscape conditions of monitored catchments are classified using hierarchical clustering analysis (lower inset; see Methods section 2.3.1). Greyed catchment names contributed to the landscape clustering analysis but were not included in this study due to insufficient flow data. Of note, though NSR Whirlpool Point is on the NSR mainstem, it is viewed in this study as a cordillera tributary as the river's origin is part of this catchment. Additional data from Shaded Relief (http://www.shadedrelief.com/world\_relief/home.html).

chernozemic soils through mixedwood forests and plains. Precipitation is low across the basin due to the Rocky Mountain rain shadow; ranging from ~600 mm yr<sup>-1</sup> in the west basin to ~350 mm yr<sup>-1</sup> in the east (Government of Alberta, 2024a). Water yields, accordingly, are highest in the western cordillera and foothills regions relative to the

lower-relief, easterly portions of the basin. The largest western tributaries of the NSR include the Brazeau, Ram, and Clearwater rivers, while smaller creeks and rivers in the central and eastern basin include the Sturgeon and Redwater rivers (Figure 1). Two reservoirs exist in the west basin on the NSR (Abraham Lake-Bighorn Dam) and the Brazeau River (Brazeau Reservoir-Brazeau Dam).

#### 2.2 Data

#### 2.2.1 Catchment structure

With a goal to classify different landscape types of monitored catchments across NSR basin (section 2.3), we used geospatial data related to the catchment structural unit (CSU) approach (Orwin et al., 2022). CSUs are a series of unique numerical identifiers derived from geospatial data of land use/landcover, surficial geology, wetlands, and slope that are known to play significant roles in how catchments respond to hydrometeorological inputs (Orwin et al., 2022). CSU codes are generated at a 400 m² pixel scale across the basin, to provide wall-to-wall coverage. For example, a pixel code of 60111 represents a dominant shrubland cover (60,000), mixed coarse-fine surface geology (100), with wetlands present (10), and a slope less than 10% (1). Further data details are provided in the supporting information and Table S1a.

#### 2.2.2 Hydrometeorology

Mean annual air temperature and total annual precipitation data were retrieved from Canadian Climate Normal meteorological stations (1981-2010) across the NSR basin (Table 1; Government of Canada, 2024). Mean annual air temperature and total annual precipitation for the water years (November 1<sup>st</sup> to October 31<sup>st</sup>) 2019 to 2022 were also collected from the same stations. Mean annual river flow data were collected from either long-term federal (Government of Canada, 2023) or provincial (Government of Alberta, 2023) monitoring stations across the NSR basin (Table 1). Historic data were collected between the water years of 1981 and 2010. Mean annual flow from water years 2019 to 2022 were also collected from the same stations.

<u>2</u>02

<u>203</u>

**Table 1** Metrics of meteorological (Canadian Climate Normals; Government of Canada, 2024), water quality (WaterSHED Monitoring Program, Long Term River Network; Emmerton et al., 2023), and flow (Government of Alberta, 2023; Government of Canada, 2023) stations of the North Saskatchewan River (NSR) basin used in this study. Paired flow indicates flow monitoring occurs at same locations as water quality sampling. Stations ordered upstream to downstream through the basin. Open-water (May to October; OW); M-semi-monthly; C-continuous.

| Meteorology              |                        |                                  |                               |                 |                      |  |  |
|--------------------------|------------------------|----------------------------------|-------------------------------|-----------------|----------------------|--|--|
| Station                  | Latitude,<br>Longitude | Monitoring frequency             | Canadian<br>Climate<br>Normal | Study<br>period | Elevation<br>(m asl) |  |  |
| Nordegg RS               | 52.500, -116.050       | С                                | 1981-2010                     | 2018-2022       | 1320.1               |  |  |
| Entwistle                | 53.600, -114.983       | С                                | 1981-2010                     | 2018-2022       | 780.3                |  |  |
| Edmonton Stony Plain     | 53.548, -114.108       | С                                | 1981-2010                     | 2018-2022       | 766.3                |  |  |
| Edmonton International A | 53.317, -113.583       | С                                | 1981-2010                     | 2018-2022       | 723.3                |  |  |
| Elk Island National Park | 53.683, -112.868       | С                                | 1981-2010                     | 2018-2022       | 716.2                |  |  |
|                          | Water Quality          |                                  |                               | Flow            |                      |  |  |
| Station                  | Latitude,<br>Longitude | Monitoring frequency (# samples) | Catchment<br>Area (km²)       | Paired flow?    | Monitoring frequency |  |  |
| NSR at Whirlpool Pt.     | 52.001, -116.471       | M (54)                           | 1,920                         | Yes             | С                    |  |  |
| NSR at Saunders Camp.    | 52.454, -115.759       | M (47)                           | 5,089                         | Noa             | С                    |  |  |
| Ram R.                   | 52.368, -115.420       | M (55)                           | 1,881                         | Yes             | С                    |  |  |
| NSR u/s Clearwater R.    | 52.348, -114.982       | M (45)                           | 7,715                         | Nob             | С                    |  |  |
| Clearwater R.            | 52.253, -114.856       | M (55)                           | 3,221                         | Yes             | С                    |  |  |
| Baptiste R.              | 52.664, -115.076       | M (53)                           | 1,358                         | Yes             | С                    |  |  |
| Nordegg R.               | 52.820, -115.513       | M (54)                           | 865                           | Yes             | С                    |  |  |
| Brazeau R.               | 52.913, -115.364       | M (51)                           | 5,658                         | Yes             | С                    |  |  |
| Rose Ck.                 | 53.052, -115.051       | M-OW (52)                        | 657                           | Yesc            | C-OW                 |  |  |
| Modeste Ck.              | 53.248, -114.706       | M-OW (40)                        | 1,178                         | Yes             | C-OW                 |  |  |
| Tomahawk Ck.             | 53.352, -114.660       | M-OW (36)                        | 190                           | Yesd            | C-OW                 |  |  |
| Strawberry Ck.           | 53.311, -114.052       | M-OW (51)                        | 589                           | Yes             | C-OW                 |  |  |
| NSR at Devon             | 53.369, -113.751       | M (50)                           | 26,362                        | Noe             | C-OW                 |  |  |
| Whitemud Ck.             | 53.484, -113.555       | M-OW (39)                        | 1,086                         | Yes             | C-OW                 |  |  |
| Sturgeon R.              | 53.833, -113.283       | M-OW (37)                        | 3,330                         | Yes             | C-OW                 |  |  |
| Redwater R.              | 53.897, -113.000       | M-OW (37)                        | 1,602                         | Yes             | C-OW                 |  |  |
| NSR at Pakan Bridge      | 53.991, -112.476       | M (51)                           | 39,333                        | Yesf            | С                    |  |  |

Notes: aNSR at Saunders Camp. uses flow from upstream NSR at Bighorn Dam (05DC010) and uses water yield from Ram R. (05DC006) to estimate additive flow from ungauged drainage areas; bNSR u/s Clearwater R. uses flow from downstream NSR near Rocky Mountain House (05DC001) and removes flow from Clearwater R. at Dovercourt (05DB006) and Prairie Ck. near Rocky Mountain House (05DB002); bNewly installed paired station in 2019 without a long-term record. Long-term record available upstream at Rose Creek at Alder Flats (05DE007); bNewly installed paired station in 2019 without long-term record. Long-term record available upstream at Tomahawk Creek at Tomahawk (05DE009); NSR at Devon uses flow from downstream NSR at Edmonton (05DF001) and removes flow from Whitemud Ck. at Edmonton (05DF009); NSR at Pakan Bridge uses paired flow after Nov. 6/20, otherwise uses flow from upstream NSR at Edmonton (05DF009), adds flow from Sturgeon R. (05EA001), Redwater R. (05EC005), and Waskatenau Ck. (05EC002) and uses water yield from Redwater R. (05EC005) to estimate additive flow from ungauged drainage areas. Station information can be found from Water Survey of Canada (Government of Canada, 2023) and Alberta River Basins (Government of Alberta, 2023).

#### 2.2.3 Water quality

Tributary monitoring occurs at the mouths of 20 WaterSHED catchments (Table 1; Figure 1) and includes: 1. flow gauging that monitors water level and estimates river flow every 15 minutes using rating curves; and 2. flow-targeted water quality grab sampling (i.e., twice-monthly monitoring during spring freshet; monthly thereafter) at 30 cm depth throughout the open water season (March–October) at all sites and monthly winter sampling (below-ice) at larger tributary stations. Water samples are collected and analyzed for general chemistry, nutrients, trace metals, contaminants, and biological measures at each station (Emmerton et al., 2023). Four mainstem stations of the NSR are monitored for water quality but are not necessarily co-located with flow stations. Therefore, flow was estimated at water quality stations using records from the nearest flow station and local tributary water yields applied to remaining ungauged catchment area between stations (Table 1; Figure 1; see supporting information). Sample collection and laboratory analysis methods under this program are the same as those under WaterSHED, though sample collections at NSR mainstem stations are at a monthly frequency only.

Only TSS and DOC were used as water quality parameters for numerical assessments, for reasons described in the following section. TSS was measured using the gravimetric approach (SM 23 2540Cm; Rice et al., 2017) with 2 µm filtration and a reported detection limit of 1 mg L<sup>-1</sup>. Colorimetry via an infrared autoanalyzer (SM 22 5310Bm; Rice et al., 2017) was used to determine DOC concentrations in water samples at a detection limit of 0.5 mg L<sup>-1</sup> after 0.45 µm filtration. All WaterSHED and NSR mainstem data are publicly available (Government of Alberta, 2024b) and are assessed for quality control and assurance (see supporting information; Laceby et al., 2022).

#### 2.3 Numerical assessments

#### 2.3.1 Catchment structure

As the CSU codes are generated in a wall-to-wall coverage across the NSR basin, numeric summaries and distribution of CSU codes can be generated within catchment areas. We used ArcGIS (v10.7.1) HydroTools and a 100 m interpolated digital elevation

model to delineate areas of each of the 20 WaterSHED catchments covering most of the NSR basin. Individual CSU codes in each catchment were summed by area to quantify a percent catchment coverage of each code. To classify different landscape types across NSR basin, we produced a matrix of CSU codes and areal percentages for each WaterSHED catchment and entered it into a hierarchical cluster analysis using Euclidean distances and Ward's approach (IBM SPSS v28). The number of clusters selected for further assessments was chosen manually to balance replication of catchments within a reduced number of clusters. These landscape types thus formed the basis for subsequent statistical assessments to understand how riverine transport of TSS and DOC were modulated by hydrometeorological conditions across differing landscape conditions across the basin.

#### 2.3.2 Hydrometeorology

To contextualize recent conditions relative to those from the past, we compared mean annual air temperatures and total annual precipitation from a meteorological station during its climate normal period (1981–2010) with mean air temperature and total precipitation from the 2019 to 2022 water years at the same station. Similarly, we compared historic mean annual flow at a given long-term flow monitoring station (1981 to 2010) to mean annual flow from the water years 2019 to 2022 at the same station. Total annual flow for each water year and each station was calculated by summing daily flows. Total annual runoff yield (mm) was then calculated by dividing the annual flows by gross catchment areas.

#### 2.3.3 Concentrations, export, and catchment yields

We chose 13 of the 20 catchment stations to include in concentration, chemical export, and catchment yield assessments since long-term flow data (and therefore reliable rating curves) were not available yet at all catchments (Table 1). Eleven of these 13 stations had paired long-term flow station records, while two new stations were included in assessments (Rose, Tomahawk) since they had long-term flow records available at upstream stations. TSS and DOC were selected as representative parameters for this study because previous work from Alberta river monitoring programs has shown notable concentration associations between TSS and several unfiltered

trace metals (e.g., Al, Co, Cr, Cu, Fe, Pb, V, Zn), while DOC concentrations showed patterns similar to total and dissolved forms of nitrogen and phosphorus, and dissolved trace contaminants (e.g., As, Fe, Hg, V; Emmerton et al., 2023). Of note, DOC is the majority of total organic carbon in these systems (Government of Alberta, 2024b).

Quantification of export (i.e., concentration x flow) in rivers typically includes assessments of parameter concentrations from grab samples collected relatively infrequently and higher frequency river flow monitoring. Over a period of several years, grab samples from a given station are collected across much of a river's hydrological and biogeochemical conditions (Figure S1). Quantification of export (mass) from rivers may be achieved using several different approaches. We used the R package loadflex (Appling et al., 2015) to estimate annual export at catchment and NSR mainstem monitoring stations using four different export models. The first model uses a rectangular interpolation approach that connects consecutive concentration measurements using a period-weighted average approach. Daily export is quantified using the product of filled concentrations and flow data on a given day. The second model is a log regression model between periodically measured concentrations and its paired flow data on the day of collection. This model is then used with the full flow data record to quantify daily export estimates. The third model is the United States Geological Survey Loadest model (Runkel et al., 2004). This approach fits several different regressions to periodic concentration data and paired flow to find the best performing regression model to fit the data. This model is then used with flow data to quantify daily export. Finally, the composite model uses a regression model (Loadest as default) and collected concentration data to generate a residual dataset between regression estimates and real data. These residuals are then used systematically to correct regression estimates to provide an improved regression that is used with flow to quantify daily export. Daily uncertainty is quantified as a standard error of prediction (SEP) for each model as outlined in Appling et al. (2015). Briefly, interpolation models use a jackknife process where interpolation models are fit to the data not including one observation and differences between the removed data point and the predicted point are quantified. For regression-based models (i.e., regression, Loadest), daily SEP is determined as the square root of the sum of coefficient uncertainty. The composite

model uses a composite parametric bootstrap approach. To produce annual water year export for each site and each model, we summed all daily export estimates for each water year and associated daily SEP via summing propagation of error rules. For errors related to unmonitored flow, we used the mean proportional error of the river(s) used to calculate unmonitored flows via their yields (see Methods). Errors related to the change in storage ( $\Delta$ S) term were determined by simple error propagation rules. Annual tributary catchment yields (mass per area) were quantified by dividing estimated export by gross catchment area for each station.

To quantify TSS and DOC export and storage (in ktonnes) of four NSR mainstem reaches between water quality monitoring stations, we used median export of the four models at all stations and a mass balance approach (after Yin et al., 2021):

$$\Delta S_{NSR} = I_{MT} + I_{UT} - (O_{NSR(DS)} - I_{NSR(US)}) + E$$
 (1)

where  $I_{MT}$  denotes reach inputs from monitored tributary catchments;  $I_{UT}$  is unmonitored tributary export to the reach using the product of catchment yields of other nearby monitored tributaries and total unmonitored land area draining to the reach;  $O_{NSR(DS)}$  represents the NSR mainstem station export at the downstream bound of the reach; and  $I_{NSR(US)}$  is the NSR mainstem station export at the upstream bound of the reach.  $\Delta S_{NSR}$  is assumed to be contributions to or subtractions from the NSR mainstem from in-channel sediment deposition (–), in-channel/bed or bank erosion (+), or additions or subtractions from riparian and floodplain regions ( $\pm$ ). The residual (E) includes groundwater or atmospheric contributions and losses of TSS or DOC, which are assumed low relative to surface runoff contributions and not considered here.

#### 2.3.4 Linear mixed models

To assess statistical differences between catchment yields of TSS and DOC across different landscape types, between water years, and between four different export models, we used linear mixed models (SPSS v28). The three fixed factors in the mixed models were water year, landscape type, and export model while the random factor was monitoring stations using an autoregressive approach:

 $\label{eq:Yieldijklm} Yield_{ijklm} = \beta_0 + \beta_1 Landscape_i + \beta_2 Year_j + \beta_3 Model_k + \beta_4 (Landscape_i \times Year_j) + \beta_5 \qquad (2) \\ (Landscape_i \times Model_k) + \beta_6 (Year_j \times Model_k) + \beta_7 (Landscape_i \times Year_j \times Model_k) + \textit{U}_l + \varepsilon_{ijklm}$ 

where Yield is the TSS or DOC yield,  $\beta_0$  is the overall intercept,  $\beta_{n+1}$  are the fixed coefficients; Year/Landscape/Model are the fixed factors,  $u_l$  is the river station random factor, and  $\epsilon_{ijklm}$  is the residual error. Daily catchment yield data (dependent variable) were strongly and positively skewed and were log-transformed to improve normality to meet linear mixed model standards (Knief and Forstmeier, 2021). Sidak multiple comparisons (Sidak, 1967) were used to assess statistical differences ( $\alpha$ =0.05) between geometric means of TSS and DOC yields by landscape type, water year, and export model. Models were run using yield data from all monitored catchments, but run again excluding the regulated Brazeau River station to remove effects of the dam and reservoir upstream of that station on sediment transport and DOC.

#### 3. Results

#### 3.1 Catchment structure

Using the CSU approach (Table S1a) and hierarchical clustering of all WaterSHED catchments, we identified four, statistically distinct tributary landscape types across the NSR basin (Figure 1; Table S1b). Cordillera catchments are dominated by steep rock/rubble land cover, exposed bedrock, bare and forested slopes, and few wetlands or anthropogenic land uses. Foothills catchments are typically forested with wetlands, have coarse and fine surface sediments, and generally have rolling topography. Forestry is the dominant land use, with less extensive footprints by oil and gas development and linear features. Mixed agriculture catchments are dominated by low-slope agricultural lands, with patches of mixedwood forest, grasslands, large lakes, and energy production footprints. Wetland coverage is low and surficial geology is mixed with coarse sediments dominating. Finally, agriculture catchments are heavily agriculturalized landscapes with coarse tills, few wetlands, limited open water, and abundant linear features relative to other classes due to proximities to urban areas and rural road grids. The NSR at Whirlpool Point flow station was the only catchment to

drain a substantial glacier and these meltwaters were delivered downstream to other NSR mainstem stations.

#### 3.2 Hydrometeorology

Regardless of geography, monitoring stations across the NSR basin experienced relatively cool air temperatures and the highest precipitation accumulations and subsequent river flows during the water years 2019 and 2020 (Figure 2). One-year standardized precipitation index mapping (Figure S2) suggested between normal and one in three-year wet conditions persisted across most of the NSR basin in 2019 but the Brazeau catchment experienced up to 1 in 25-year wet conditions. In 2020, the NSR basin experienced widespread one in three-year wet conditions with patches of up to 1 in 12-year wet conditions. In contrast, 2021 was characterized by extreme heat and drought highlighted by a widely-reported atmospheric blocking (heat dome) event over much of western Canada. Consequently, 1 in 25-year, up to 1 in 100-year dry conditions prevailed across the NSR basin, particularly in foothills and downstream plains regions. Meteorological conditions across most of the basin were near-normal in 2022.

The cordillera NSR at Whirlpool Point and the regulated Brazeau River stations were the largest tributary contributors of flow to the NSR of monitored stations, each contributing ~25% of all flow at the NSR–Edmonton station (Figure 1) between water years 2019 and 2022 (Figure 2). Other cordillera rivers including the Ram and Clearwater were the next notable contributors of river water to the NSR, while downstream foothills and plains rivers were less important for flow. Catchment water yields (runoff) also generally followed this downstream progression (Table S2).

Between years (Figure 2; Table S2), the glacially-sourced NSR at Whirlpool Point had the lowest variability in mean annual flow (50<sup>th</sup>–52<sup>nd</sup> percentile of historic record) and was unique across the network as it experienced relatively high flows during much of the hot and dry 2021 summer. The regulated Brazeau River produced stable but high flows between 2019 and 2022 (57<sup>th</sup>–64<sup>th</sup>), particularly during a very wet 2019 in the catchment. Other cordillera rivers showed lower relative flows and higher interannual

**Figure 2** Canadian climate normals for mean annual air temperature (upper panel bars) and total annual precipitation (centre panel bars) and mean annual river flow (lower panel bars) from monitoring stations across the North Saskatchewan River (NSR) basin (1981-2010; ±1 standard error). Annual metrics from 2019 to 2022 water years are overlain as symbols. All stations are ordered west to east through the basin and mainstem stations include the prefix "NSR" in their labels and (M) labels denote meteorological stations. \*Station historic flow data from longer-term upstream stations (Rose Creek at Alder Flats; Tomahawk Creek at Tomahawk).

flow variabilities (44<sup>th</sup>–62<sup>nd</sup>). Both foothills (39<sup>th</sup>–69<sup>th</sup>) and plains rivers (38<sup>th</sup>–85<sup>th</sup>) were even more variable in flow conditions between years. Of note, cordillera rivers Ram and Clearwater showed an overall annual flow condition that was near normal in 2022 despite experiencing isolated storm events in early June 2022 that increased their flows substantially (i.e., 99<sup>th</sup> percentile of flows) for a short period. Flow at the NSR at Edmonton was mostly above its 115-year upper quartile of flow in 2020 but mostly below its lower quartile in 2021. Other years tended toward historical median flow. Total flow at the station ordered highest to lowest was 9.4 km³ (2020), 7.9 km³ (2019), 6.5 km³ (2022), then 5.7 km³ (2021).

#### 3.3 TSS and DOC concentrations

Downstream through the NSR basin, TSS concentrations in catchments showed a step-change pattern (Figure 3). Relatively low, but variable concentrations occurred in the cordillera (mean of all samples  $\pm$  standard error:  $64.0 \pm 23.6$  mg L<sup>-1</sup>) and foothills (25.0  $\pm$  11.1 mg L<sup>-1</sup>) regions and higher concentrations in the agriculturalized plains (mixed agriculture:  $83.7 \pm 24.0$  mg L<sup>-1</sup>; agriculture:  $100.6 \pm 20.8$  mg L<sup>-1</sup>). Concentrations of TSS were often highest in the wetter 2019 and 2020 water years compared to the drier 2021 and 2022 years, but differences occurred depending on the catchment. In geographical agreement with the catchments, the NSR mainstem progressed from 14.0  $\pm$  2.7 mg L<sup>-1</sup> of TSS in the two upper basin stations to a more variable  $102.7 \pm 32.5$  mg L<sup>-1</sup> at the downstream plains stations. The flow regulated NSR mainstem stations reported more consistent concentrations between years compared to catchments.

Downstream concentration changes of DOC in catchments also showed a step-change, but between the cordillera catchments  $(2.1 \pm 0.1 \text{ mg L}^{-1})$  and all other stations (range:  $7.0 - 17.7 \text{ mg L}^{-1}$ ). NSR mainstem stations mirrored this pattern (upper stations:  $0.9 \pm 0.1 \text{ mg L}^{-1}$ ; lower stations:  $3.5 \pm 0.3 \text{ mg L}^{-1}$ ). Again, wetter years produced greater DOC concentrations compared to drier years. Variability in DOC concentrations between all sites and water years was lower compared to TSS (Figure 3).

With differing degrees of association, concentrations of TSS in most catchment and NSR mainstem stations coupled positively with flow and thus produced well-performing concentration—flow models (Table S3, Figure S3). An exception to this pattern occurred downstream of the dam-influenced NSR-Saunders station, whereas concentration—flow

**Figure 3** Concentration boxplots (10-25-50-75-90<sup>th</sup> percentile boxes/whiskers; dot outliers) of total suspended sediment (TSS, upper panel) and dissolved organic carbon (DOC, lower panel) by water year for all tributary catchment (filled boxes) and North Saskatchewan River (NSR) mainstem (open boxes) monitoring stations ordered west to east across the basin. Tributary catchments are classified by landscape type (see Methods). Dashed red line indicates the analytical reporting limit for each water quality parameter with the few samples below that line reported at half the detection limit.

associations from the flow regulated Brazeau River were stronger. DOC concentration—flow associations were also robust across the basin, though Redwater, Tomahawk, and NSR-Whirlpool Point catchments showed weaker associations compared to others.

#### 3.4 TSS and DOC catchment yields

Catchment yields of TSS throughout the NSR basin were highly variable between water years (Table S4). All TSS yields ranged from <1 to ~325 tonne km<sup>-2</sup> yr<sup>-1</sup> with highest observed yields from mixed agriculture, agriculture, and foothills catchments in 2020, and cordillera catchments in 2022. Punctuating this variability, TSS yields from the highest sediment yielding catchments in 2020 were approximately 100 to nearly 300 times larger than yields in the 2021 dry year. Only catchments that were regulated (i.e., Brazeau) or drained glacially influenced landscapes (NSR-Whirlpool Point) showed relatively consistent TSS yields between water years. DOC catchment yields were much smaller than TSS yields ranging from <1 to only 7 tonne km<sup>-2</sup> yr<sup>-1</sup> and yields were highest through the central portion of the NSR basin (Table S5). However, the variability of DOC yields between years and catchments was much lower than for TSS. For NSR mainstem stations, both TSS and DOC yields were least variable between water years in the headwaters and more variable in the plains regions.

Cordillera TSS catchment yields were statistically larger than those from mixed agriculture and agriculture landscapes (Figure 4; Table S6). Foothills yields were not different than those of other landscape types. These patterns were true for TSS regardless of the differences in hydrometeorological conditions between water years or if the regulated Brazeau catchment was included or not. Across all landscapes and export model types, catchment yields of TSS were highest in the wet 2019 and 2020 years and both were statistically larger than yields in 2021 and 2022 (i.e., 2019=2020>2022>2021). We observed increases in TSS yields that far outweighed increases in flow between water years, particularly at certain foothills (e.g., Nordegg, Rose), mixed agriculture (i.e., Modeste, Tomahawk), and agriculture (e.g., Strawberry, Redwater) catchments (Table 2). For example, the Nordegg River in the foothills reported TSS yields that were ~200 times higher in 2020 compared to 2021, despite

**Table 2** Ratios of annual (water years 2019 – 2022) maximum to minimum runoff (from Table S2), total suspended sediment (TSS) yield (from Table S4), and dissolved organic carbon (DOC) yield (from Table S5) for all tributaries and North Saskatchewan River (NSR) mainstem (emphasized) sites. C-cordillera; F-foothills; MA-mixed agriculture; A-agriculture.

| Station                  | MAX:MIN<br>Runoff | MAX:MIN<br>TSS Yield | MAX:MIN<br>DOC Yield |
|--------------------------|-------------------|----------------------|----------------------|
| NSR at Whirlpool Pt. (C) | 1                 | 3                    | 1                    |
| NSR at Saunders Camp.    | 1                 | 1                    | 1                    |
| Ram R. (C)               | 2                 | 18                   | 3                    |
| NSR u/s Clearwater R.    | 1                 | 3                    | 2                    |
| Clearwater R. (C)        | 2                 | 24                   | 3                    |
| Baptiste R. (F)          | 4                 | 51                   | 5                    |
| Nordegg R. (F)           | 3                 | 193                  | 6                    |
| Brazeau R. (C)           | 2                 | 5                    | 4                    |
| Rose Ck. (F)             | 10                | 289                  | 16                   |
| Modeste Ck. (MA)         | 13                | 153                  | 17                   |
| Tomahawk Ck. (MA)        | 57                | 481                  | 103                  |
| Strawberry Ck. (A)       | 8                 | 107                  | 15                   |
| NSR at Devon             | 2                 | 23                   | 5                    |
| Whitemud Ck. (A)         | 10                | 33                   | 17                   |
| Sturgeon R. (MA)         | 13                | 59                   | 16                   |
| Redwater R. (A)          | 32                | 108                  | 49                   |
| NSR at Pakan Bridge      | 2                 | 51                   | 5                    |

runoff being only three times higher in 2020. Similarly, some cordillera rivers (i.e., Ram, Clearwater) reported annual TSS yields 18 to 25 times higher in 2022 compared to 2021 with runoff only 1.3 times higher in 2022. TSS catchment yields were highly variable (even magnitudes apart) between export models despite reporting the same patterns between landscapes and water years (Figure 4), similar to other work (Kamrath et al., 2023). Across the four different TSS export models, we found statistical differences in yields between each model except for the Loadest and composite models ( $\alpha$ =0.05; Table S6).

For DOC catchment yields, all years were statistically different following the descending order 2019, 2020, 2022, and 2021 (Figure 4; Table S7). Though yields showed the same patterns between landscape types as TSS, there were discrepancies between years. DOC yields were not statistically different between landscape types in the wetter 2019 and 2020 years, but cordillera DOC yields were higher in the drier 2021 and 2022 years compared to mixed agriculture and agriculture landscapes. Removal of the Brazeau station did not affect these patterns. River stations across the NSR basin

**Figure 4** Total suspended sediment (TSS; upper panel) and dissolved organic carbon (DOC; lower panel) catchment yields (filled circles) from water years 2019 to 2022 from tributary catchments of the North Saskatchewan River basin ordered upstream to downstream. Yield variability of each site reflects results from different loadflex export models (see Methods). Linear mixed model results (right of graphs) assess statistical differences between landscapes, water years, and their combination across all export models. Different letters within a given category (i.e., landscape, water year) denote statistical differences as determined by the linear mixed model.

reported interannual maximum to minimum DOC yield ratios of 1 to 103 (mean: 16) compared to 1-57 (mean: 10) for runoff, which were much closer together relative to TSS (1-481; mean 94; Table 2). DOC yield estimates were much less variable between export models compared to TSS results (Figure 4). DOC export models reported that yields computed by composite and regression approaches, and interpolation and Loadest approaches, were statistically similar (Table S7).

### 3.5 TSS and DOC export

Export of TSS through the catchment and NSR mainstem network displayed sharp interannual contrasts (Figure 5; Table S8). Across all years, catchment (and unmonitored land) export of TSS was highest in the near-normal 2022 (median of models ± SE: 2,239 ± 962 ktonne), followed by the wetter years 2020 (2,183 ± 669 ktonne) and 2019 (627 ± 216 ktonnes), then the drought 2021 year (316 ± 212 ktonne). Though most tributaries showed their highest exports in the wet 2020 year, extremely high sediment export from the cordillera Ram and Clearwater rivers during substantial, short-term rainfall events in June dominated tributary sediment export in 2022.

Storage of sediment (Figure 5) in the upstream NSR-Saunders reach (194 – 788 ktonnes yr<sup>-1</sup>, 2019 to 2022) was closely aligned with incoming delivery of sediment from catchments (219 – 807 ktonnes yr<sup>-1</sup>). This pattern was true even during substantial glacier melt-related transport of sediments during the 2021 heat dome event. The behaviour of Abraham reservoir upstream of the Bighorn Dam as a major sediment sink is consistent with this overall relationship between inputs and storage.

The NSR-upstream Clearwater reach showed variable inputs of sediments from catchments each year (31 – 564 ktonne yr<sup>-1</sup>), with highest export in 2022. From 2019 to 2020, there was storage of sediment in the river (40 – 48 ktonne yr<sup>-1</sup>) before a modest release of sediment in 2021 (15 ktonne yr<sup>-1</sup>). In 2022, sediment contributed by the Ram River and unmonitored regions after June storms (564 ktonne yr<sup>-1</sup>) was mostly stored in this reach of the NSR (428 ktonne yr<sup>-1</sup>).

Through the lower-slope NSR-Devon reach in 2019 to 2021, addition of sediment to the river from stores (48 – 1,719 ktonne yr<sup>-1</sup>) outpaced catchment sediment additions by 1.2 to 2.5 times, meaning a majority of suspended sediment in this reach of the NSR

**Figure 5** Total suspended sediment (TSS) export arrow diagrams of the North Saskatchewan River (NSR) basin for the water years (November to October) of 2019 to 2022. Tributary and storage exchanges are represented by horizontal arrows and NSR mainstem stations and reaches are shown as vertical arrows. Alternating shading delineates NSR mainstem reaches. Relative station spatial separations not to scale. Medians of all export models were used to construct mass arrows.

was sourced from areas within and adjacent to the river itself. The high-flow year of 2020 resulted in major additions of sediment to the river from in-river sources. In 2022, contributions of sediment from Clearwater R. and other tributaries were stored in the reach (542 ktonne yr<sup>-1</sup>), in contrast to previous years.

Finally, through the NSR-Pakan reach, the banks and channels of the NSR itself contributed 734 and 53 ktonne yr<sup>-1</sup> of sediment in 2020 and 2022, respectively, while 2019 reported 78 ktonne yr<sup>-1</sup> of sediment storage through the reach while 2021 had storage of 68 ktonne yr<sup>-1</sup>. Across all NSR mainstem stations, sediment storage occurred mostly in the upper two station reaches (7/8 occurrences) compared to sediment losses in the two downstream reaches (5/8 occurrences). Overall, catchment export relative to the furthest downstream mainstem station (NSR at Pakan) was 49-61% in the wet 2019 and 2020 years up to 245% (2022) and 360% (2021) during the normal to drier years.

In contrast to the highly variable and active storage terms year to year for TSS, export of DOC downstream through the NSR basin showed more consistent patterns (Figure 6; Table S9). In upper reaches, export of DOC from tributaries was relatively low but consistent between years delivering 1.4 to 6.1 ktonne yr¹ following the interannual pattern 2020>2019>2022>2021. Consistent storage of DOC in the dam-influenced NSR-Saunders reach each water year mirrored TSS patterns. DOC was stored through the NSR-upstream Clearwater reach but storage was nearer to net zero compared to the NSR-Saunders reach. Through the NSR-Devon reach, DOC was stored except during the wet 2020 season when DOC was added to the river. The NSR-Pakan reach reported near zero storage in 2019 before consistent DOC addition to the river (0.5 – 3.1 ktonne yr¹) from 2020 to 2022. Overall, cycling of DOC between the NSR mainstem and its stores within and adjacent to the river was less consequential than for TSS, resulting in relatively consistent tributary contributions of DOC relative to NSR at Pakan station of between 97 and 130%.

#### 4. Discussion

#### 4.1 Sediment mobility driven by site-specific precipitation and erosion thresholds

Numerous studies have demonstrated that different catchment characteristics can influence downstream water quality (Allan, 2004; Lintern et al., 2018b; Shi et al., 2024). These landscape to water quality connections require a source of landscape material in the catchment, precipitation and runoff, entrainment or dissolution of material, and downstream delivery. Typically, we expect steep cordillera regions to have colluvial sediment stores of large grain sizes that produce fewer suspended sediments in rivers

**Figure 6** Dissolved organic carbon (DOC) export arrow diagrams of the North Saskatchewan River (NSR) basin for the water years (November to October) of 2019 to 2022. Tributary and storage exchanges are represented by horizontal arrows and NSR mainstem stations and reaches are shown as vertical arrows. Alternating shading delineates NSR mainstem reaches. Relative station spatial separations not to scale. Medians of all export models were used to construct mass arrows.

(Sinha and Friend, 1994; Townsend-Small et al., 2008) compared to thicker, unvegetated soils of erosion-prone agricultural plains and urban environments (Giri, 2021). Forested landscapes often have moderate soil stores resistant to erosion due to vegetation cover (Veldkamp et al., 2020). This pattern is reflected in the NSR basin by differences observed between lower concentration cordillera regions and those downstream (Figure 3). However, sediment yields were highest from the wetter

cordillera areas, pointing to precipitation and runoff controlling sediment mobility across the basin (i.e., transport limited; Su et al., 2023). This finding is supported by strong and positive TSS concentration-flow (C-Q) relationships at most stations and suggests there is ample sediment supply across the basin but hydrological energy is required to mobilize it (Creed et al., 2015), as observed elsewhere in Alberta (Loiselle et al., 2020).

Statistical differences in TSS yields and variability in sediment export between water years (Figures 4, 5) reflected the inconsistency in precipitation and runoff conditions observed across the NSR basin during the study. Similarly, TSS yield patterns across landscape types were the same across all water years, also suggesting hydrometeorological conditions were the dominant control on sediment yields. During the open water period in central Alberta (May-October), rainfall frequency and intensity is related to hourly-scale, patchy convective rainfall events and daily-scale low pressure systems from the west and south (Vickers et al., 2001). These drivers contribute to a mosaic of moisture conditions across the basin (Figure S2). These episodic processes lead to shorter-term hydrologic connections across landscapes and subsequent erosion that may mobilize a substantial portion of a river's annual export of material. In between events, or during dry years of this continental basin, evapotranspiration is strong, leading to desiccation of landscapes, prolonged lack of hydrologic connectivity, and higher groundwater contributions to rivers relative to surface flow (Tetzlaff et al., 2024). This juxtaposition of conditions at least partially drives the variability of sediment yields and export between years and between rivers across the basin. However, rainfall and runoff accumulation does not fully explain TSS yield variability in space and time.

Though transport-limitation and hydrometeorology drive sediment yields in both catchments and the NSR mainstem, we observed non-linear (i.e., threshold) response of sediment yields in selected tributaries relative to changes in flow, which is commonly observed in rivers (Vercruysse et al., 2017). This is consistent with rapid connection, disconnection, and reconnection events that occur between runoff and sediment stores in catchments (Fryirs, 2013). The striking sediment yield increases in 2020 in selected foothills and plains catchments and in 2022 in selected cordillera catchments demonstrates the variability (i.e., site specificity) of this threshold behavior. Threshold response also helps explain wide variability in export model results (Figure 4) as

extreme high events perturb model fits at high flows and concentrations of C-Q associations (Asselman, 2000). Further, our nested monitoring program allowed us to observe how alluvial sediment stores in major rivers interact with flow. Downstream through the NSR mainstem reaches, we observed mostly loss of sediment stores due to erosion within the NSR (2019-2021) before a reversal to sediment storage (2022; Figure 5). In 2020, a flow threshold was crossed where bank and bed erosion were more intense in the NSR-Devon reach compared to other reaches and years. These results demonstrate how mainstem flows begin to erode and entrain in-channel and marginal sediment stores and mobilize sediment downstream as an erratic "conveyor belt" (Fryirs, 2013). This behavior is particularly robust in headwater basins that transition from mountainous, highland-type landscapes to low-relief plains regions. This type of dynamic sediment transport is also typical across deglaciated landscapes like the NSR basin.

Accumulated glacial tills are often unstable and erodible and serve as a persistent source of sediment to rivers, elevating sediment yields in an effect described as paraglacial sedimentation (Ballantyne, 2002; Church and Ryder, 1972). Downstream delivery of glacial derived sediment reduces sharply over time as glacial ice retreats and easily mobilized sediment is either exhausted or becomes inaccessible to the fluvial system. Ultimately sediment yields decline, marking the end of the paraglacial period. However, fluvial reworking can create secondary stores of glaciogenic sediments downstream and when coupled with extra-channel stores, can become ongoing sources of sediment for much longer periods (Ballantyne, 2002; Orwin and Smart, 2004). This sediment may then mobilize under certain conditions when kinetic energy thresholds are crossed, typically during isolated, intense, short-term hydrological events. These types of threshold conditions are now appearing in other northern landscapes that are experiencing enhanced terrestrial-aquatic change in particulate and dissolved material export due climatic shifts (Beel et al., 2020).

# 4.2 Dissolved organic carbon mobility closely reflects broad hydrologic connectivity

Yields and export of DOC across the NSR basin generally showed similar patterns to TSS but with some notable differences. Riverine DOC is sourced from upstream

hydrological flushing of organic soil horizons, wetlands, and riparian areas (Neary et al., 2009). From a DOC source perspective, weaker cordilleran C-Q associations (Table S3) and sharp increasing concentrations out of the mountains, compared to TSS, suggest weaker transport-limitation toward neutral or source-limitation behavior (Creed et al., 2015; Su et al., 2023). Though sediment is sourced in cordillera areas either through glacial inputs, colluvial processes, or secondary scouring of channels and beds, Canada's Rocky Mountains have typically sporadic and thin organic soil and wetland cover in valleys (Hoffmann et al., 2014). The cordillera NSR-Whirlpool station, for example, showed a nearly flat C-Q association, meaning increasing flow in rivers in this relatively wet region did not result in more incoming DOC from upstream catchments. Other cordillera rivers had stronger C-Q associations (i.e., larger coefficients of determination; Table S3) but did not produce high concentrations of DOC, likely owing to thin soils and marginal wetland coverage in downstream portions of their catchments. Foothills regions have some wetland coverage and are underlain by more developed luvisolic soils with shallow organic layers and thus have greater organic matter stores compared to Rocky Mountain catchments. Further downstream, deeper chernozemic soils of the agriculturalized plains have very large stores of organic matter. However, similar to TSS, DOC yields were statistically highest from wetter cordillera catchments and lowest in those of drier agricultural landscapes despite clear differences in organic matter stores. Again, these results point to hydrometeorology playing the dominant role in DOC mobility downstream, but with a notable difference.

DOC yields across the basin, relative to TSS, reported less variability between years for a given river, between rivers of a specific landscape type, and between rivers across landscape types (Figure 4). DOC yields between landscapes were also different for wetter (2019 and 2020) versus drier (2021 and 2022) conditions. Maximum to minimum DOC yield ratios were similar to patterns reported for runoff (Table 2). These results point to straightforward hydrologic connectivity driving DOC yields across the basin compared to the erosive threshold behavior observed for TSS yields that contributed to high variability within landscape types. Rain and runoff increase hydrologic contact with shallow organic soils and wetlands, through runoff and interflow, resulting in more DOC delivery to rivers (Neary et al., 2009; Ward et al., 2017). The

pulse-shunt process concept (Raymond et al., 2016) similarly outlines how hydrologically active years in catchments flush organic matter from soils and transport it downstream quickly and efficiently to higher order rivers. This process can result in hydrologic connection to isolated sources of organic material across the landscape before disconnection during drier years. This process also requires contact and gravity flushing, rather than extra energy required to entrain particulate material. This concept also demonstrates why model performance was better for DOC than TSS as flushing produces less variability than erosion, particularly at the high end of C-Q associations.

Through the NSR mainstem, there was little evidence in most years that riparian areas, or other sources of DOC within and adjacent to the mainstem, influenced downstream DOC mobility (Figure 6). Rather, the NSR was more of a conduit of DOC sourced from tributary catchments. Only in the years when wet conditions were widespread (2020) did we observe likely riparian additions of DOC from regions adjacent to the NSR mainstem, though these inputs were still inconsequential.

Ultimately, in support of our hypotheses, interannual differences in hydrometeorological conditions controlled both TSS and DOC mobility through the NSR basin, but via differing processes (i.e., threshold erosion versus simple flushing). However, we did not expect that inter-catchment variability of TSS and DOC yields within a given landscape type would blur differences between obviously different landscape types as determined statistically using high-resolution geospatial datasets. This, therefore, points to the highly variable nature of dry, continental climates such as those in Alberta, the ongoing legacy of past glaciation on sediment availability, and may partially explain why landscape conditions here show a less measurable role in mobilization of river material, compared to other locations (Lintern et al., 2018a).

#### 4.3 Implications for water users within a changing river

The eastern slopes of the Canadian Rocky Mountains are a critical "water tower" for approximately 5 million people and Canada's centre of agricultural activity. This region is also experiencing substantial changes in climate and river flow related to changes in atmospheric circulation, glacial loss, seasonal shifts, and other hydrometeorological factors (Anderson and Radić, 2020; Moore et al., 2009; Newton et al., 2021; Rood et al., 2008). Climate modelling of conditions across the NSR basin for 2061–80 using data

from Canada's Changing Climate Report predicts mostly unidirectional increases in precipitation and temperature across most seasons (Figure S4; Bush and Lemmen, 2019). For example, across the basin, precipitation is expected to increase approximately 5–25% relative to reference conditions (1986–2005) during winter and open-water periods. Over the same period, seasonal air temperatures are expected to increase 3.5 to 5.5 °C throughout the basin, potentially inducing greater humidity and more intense convective and cyclonic rainfalls during the open water season. This heating may also cause more intense droughts when atmospheric cycles like El Niño induce drier conditions across western Canada (Asong et al., 2018).

Assuming similar river and catchment processes in the future, our results suggest that the NSR basin may experience increased mobilization of TSS and DOC in tributaries and the NSR mainstem in response to a wetting climate. A warming climate may also increase glacial runoff and associated glacial sediment mobilization (Staniszewska et al., 2021). For example, the relatively hot conditions that the NSR-Whirlpool catchment endured in 2021 resulted in relatively greater storage of water and sediment in the Bighorn Dam. Given the controlling factors of rainfall location, intensity, and specific catchment conditions on TSS mobility, increasing precipitation and river flow may create more turbid river systems whose conditions and behaviour are difficult to predict spatially and temporally. DOC mobilization would be expected to be more predictable based on open water season rainfall accumulations. Regardless, mobilization of more turbid and organic-rich waters may create issues for aquatic ecosystem health and human water users. Prolonged turbidity is a known stressor on ecosystem health due to sediment burial and scouring processes that impact periphyton, benthic invertebrates, and fish health and reproduction (Wood and Armitage, 1997). Sharp changes in DOC concentrations and character can upset heat and carbon cycling of local ecosystems (Spencer et al., 2012). From a human use perspective, turbidity and organic matter can provide substantial challenges for the costeffective production of treated drinking water (Price and Heberling, 2018). Though turbidity is effectively reduced in conventional treatment methods, higher turbidity requires more alum additions (and therefore higher cost) to settle particles (Farhaoui and Derraz, 2016). Similarly, high DOC in raw river water can require additions of

powdered activated carbon to avoid development of potentially hazardous disinfection byproducts in treated water (Delpla et al., 2009); again at higher costs.

In terms of managing human-related pollution effluents into rivers, many jurisdictions use a total maximum daily load (TMDL) approach to ensure effluent additions do not stress aquatic ecosystems. TMDLs integrate data from point and non-point sources of pollution in a river system to calculate how much of a concerning substance can be delivered to river systems but still meet water quality guidelines (USEPA, 2024). Turbidity and DOC can be important hosts binding metals and other contaminants of concern, including mercury, within their matrices (Delpla et al., 2009). Increases in natural background loading of these materials could have implications for the quality, release frequency, and volume of industrial and municipal effluents delivered to the NSR in the future. Similarly, climate change is expected to intensify wildfire activity across northern landscapes, potentially delivering greater amounts and different forms of material downstream with potential implications for TMDLs, and water users in general (Goode et al., 2012; Hanes et al., 2019).

Considering the potential for greater delivery of TSS and DOC downstream through the NSR and associated challenges for water users, it is important to contextualize these results from a source water protection perspective. In the NSR, source water protection narratives (EPCOR, 2020) focus on the cordillera and foothills regions of the basin, where most of the water in the NSR at Edmonton is sourced (Government of Canada, 2023; Newton and Taube, 2023) and where future land use changes are anticipated to occur. Currently, foothills and cordillera regions are less impacted by human activities relative to downstream agriculturalized plains regions (Alberta Biodiversity Monitoring Institute, 2016) and contribute water low in sediment and dissolved organic carbon to the NSR. Climate projections across the NSR basin expect weaker precipitation accumulation increases in the mountains and foothills relative to the agriculturalized plains (Bush and Lemmen, 2019) and long-term loss of glacial meltwater inputs to the NSR (Anderson and Radić, 2020). Coupled with expected future land use pressures in these regions means that it will be critical for land use decisions in the water-producing regions of the NSR be made in an informed manner, guided by data, and maintains ecosystem services (Palmer et al., 2009).

#### 5. Conclusion

Using our multi-annual, representative monitoring design for the NSR basin tributaries, we showed that catchment yields of TSS and DOC have significant variability not only between catchments of different structural/landscape conditions, but also between those that are similar. This demonstrated the utility of replication of river monitoring even in landscapes considered to have similar landscape conditions or human impacts. At the same time, interannual variability was a strong driver of TSS and DOC yields at all rivers and landscape types, suggesting hydrometeorology was a stronger influence on TSS and DOC mobilization downstream than landscape condition in the NSR basin. At the basin scale, TSS and DOC yields showed similar patterns between landscapes and years, supporting our hypothesis and further highlighting the importance of hydrometeorology in controlling the conditions of tributaries in this basin.

When integrating a major river in a nested monitoring design, we demonstrated the importance of channels, banks, and riparian areas of major rivers in controlling the amount of TSS and DOC stored in and delivered from a basin. This suggests that landscape conditions may not fully explain the water quality of major rivers, nor fully control the mobilization of terrestrial matter, and demonstrating the importance of monitoring major river systems in a nested manner. In support of our second hypothesis, the degree of in-channel contributions to basin export was greater during wet periods than dry and highlights how active major rivers are in the storage and transport of terrestrial material.

The NSR in western Canada supplies water to over 1.5 million Canadians and is experiencing change due to continued loss of glacial meltwater inputs, greater heating and rainfall accumulations, and increasing human populations. Our study suggests that NSR waters are expected to become more turbid and enriched in dissolved organic material under this climate regime and will likely cause challenges for water users (e.g., drinking water utilities), river management strategies, and aquatic biota. However, broader application of monitoring approaches that capture landscape heterogeneity, replicate monitoring of diverse catchment conditions, and include entire river basins in an integrated fashion will provide improved data and understanding of river systems and ultimately better decisions when managing water resources.

#### 793 Data availability statement:

- All data presented in this manuscript are publicly available for download from: 1. Government of Alberta
- water quality portal (<a href="https://environment.extranet.gov.ab.ca/apps/WaterQuality/dataportal/">https://environment.extranet.gov.ab.ca/apps/WaterQuality/dataportal/</a>) or by request
- (swq.requests@gov.ab.ca); 2. Water Survey of Canada hydrometric data
- (https://www.canada.ca/en/environment-climate-change/services/water-
- <u>overview/quantity/monitoring/survey.html</u>); 3. Alberta River Basins portal (<u>https://rivers.alberta.ca/</u>); and
- 4. Environment and Climate Change Canada meteorological data
- (https://climate.weather.gc.ca/index\_e.html).

#### CRediT Author contributions:

- -CE Conceptualization, Project administration, Formal analysis, Visualization, Writing-original draft,
- Writing-review & editing (orcid: 0000-0001-9511-9191) <a href="mailto:craig.emmerton@gov.ab.ca">craig.emmerton@gov.ab.ca</a>
- -JO Conceptualization, Funding acquisition, Project administration, Writing-review & editing (orcid: 0000-0001-5683-4723) john.orwin@gov.ab.ca
- -CB Conceptualization, Project administration, Writing-review & editing (orcid 0000-0001-5966-3202)
- <u>cristinabuendia@gencat.cat</u>
- -MC Conceptualization, Project administration, Funding, Writing-review & editing
- <u>mchristensen@epcor.com</u>
- -JG Formal analysis, Writing-review & editing (orcid: 0000-0001-5243-6891) jennifer.graydon@gov.ab.ca
- -BJ Investigation, Project administration <a href="mailto:brian.jackson@gov.ab.ca">brian.jackson@gov.ab.ca</a>
- -EM Investigation, Project administration <a href="mailto:elynne.murray@gov.ab.ca">elynne.murray@gov.ab.ca</a>
- -SN Project administration, Funding <a href="mailto:sneufeld@epcor.com">sneufeld@epcor.com</a>
- -BN Formal analysis, Writing-review & editing (orcid: 0000-0001-6554-6782) brandi.newton@gov.ab.ca
- -RO Investigation, Project administration <a href="mailto:ryan.ozipko@gov.ab.ca">ryan.ozipko@gov.ab.ca</a>
- -RP Investigation, Project administration <a href="mailto:rick.pickering@gov.ab.ca">rick.pickering@gov.ab.ca</a>
- -NT Formal analysis, Writing-review & editing (orcid: 0000-0003-3567-7508) nadine.taube@gov.ab.ca
- -CW Investigation, Project administration chris.ware@gov.ab.ca

- The authors of this manuscript have no conflicts of interest or competing interests with
- this work from an affiliations or funding perspective.

## 6. Acknowledgements

- We would like to thank the many hydrometric, meteorological, and water quality
- field technicians from the Government of Alberta and Government of Canada who aided
- in the collection of data for river flow and water quality for stations included in this work.
- We also are indebted to Government of Alberta data management staff including Chris
- Rickard, Lisa Reinbolt, and Jenny Pham who organized and validated data. We thank
- Dr. Bill Donahue and Dr. Fred Wrona for their work helping to establish the WaterSHED
- monitoring program. Analytical funding for this work was provided by Epcor Water
- Services for WaterSHED stations, and by the Government of Alberta for NSR mainstem
- stations.

- Alberta Biodiversity Monitoring Institute: Wall-to-Wall Human Footprint Inventory Year 2016, 2016.
- Allan, J. D.: Landscapes and riverscapes: The influence of land use on stream ecosystems, Annu. Rev. Ecol. Evol. Syst., 35, 257–284, https://doi.org/10.1146/annurev.ecolsys.35.120202.110122, 2004.
  - Anderson, S. and Radić, V.: Identification of local water resource vulnerability to rapid deglaciation in Alberta, Nat. Clim. Chang., 10, 933–938, https://doi.org/10.1038/s41558-020-0863-4, 2020.
  - Appling, A. P., Leon, M. C., and McDowell, W. H.: Reducing bias and quantifying uncertainty in watershed flux estimates: the R package loadflex, Ecosphere, 6, 1–25, https://doi.org/10.1890/ES14-00517.1, 2015.
  - Asong, Z. E., Wheater, H. S., Bonsal, B., Razavi, S., and Kurkute, S.: Historical drought patterns over Canada and their teleconnections with large-scale climate signals, Hydrol. Earth Syst. Sci., 22, 3105–3124, https://doi.org/10.5194/hess-22-3105-2018, 2018.
  - Asselman, N. E. M.: Fitting and interpretation of sediment rating curves, J. Hydrol., 234, 228–248, https://doi.org/10.1016/S0022-1694(00)00253-5, 2000.
  - Ballantyne, C. K.: A general model of paraglacial landscape response, 3, 371–376, 2002.
  - Beel, C. R., Lamoureux, S. F., Orwin, J. F., Pope, M. A., Lafrenière, M. J., and Scott, N. A.: Differential impact of thermal and physical permafrost disturbances on High Arctic dissolved and particulate fluvial fluxes, Sci. Rep., 10, 1–13, https://doi.org/10.1038/s41598-020-68824-3, 2020.
  - Brandt, J. P., Flannigan, M. D., Maynard, D. G., Thompson, I. D., and Volney, W. J. A.: An introduction to Canada's boreal zone: Ecosystem processes, health, sustainability, and environmental issues1, Environ. Rev., 21, 207–226, https://doi.org/10.1139/er-2013-0040, 2013.
    - Bush, E. and Lemmen, D. .: Canada's Changing Climate Report, 444 pp., 2019.
    - Chalov, S. and Ivanov, V.: Catchment and in-channel sources in three large Eurasian Arctic rivers: Combining monitoring, remote sensing and modelling data to construct Ob', Yenisey and Lena rivers sediment budget, Catena, 230, 107212, https://doi.org/10.1016/j.catena.2023.107212, 2023.
    - Church, M. and Ryder, J.: Paraglacial sedimentation: a consideration of fluvial processes conditions by glaciation., Geol. Soc. Am. Bull., 83, 3059–3072, 1972.
- Creed, I. F., McKnight, D. M., Pellerin, B. A., Green, M. B., Bergamaschi, B. A., Aiken, G. R., Burns, D. A., Findlay, S. E. G., Shanley, J. B., Striegl, R. G., Aulenbach, B. T., Clow, D. W., Laudon, H., McGlynn, B. L., McGuire, K. J., Smith, R. A., and Stackpoole, S. M.: The river as a chemostat: Fresh perspectives on dissolved organic matter flowing down the river continuum, Can. J. Fish. Aquat. Sci., 72, 1272–1285, https://doi.org/10.1139/cjfas-2014-0400, 2015.
- Delpla, I., Jung, A. V., Baures, E., Clement, M., and Thomas, O.: Impacts of climate change on surface water quality in relation to drinking water production, Environ. Int., 35, 1225–1233, https://doi.org/10.1016/j.envint.2009.07.001, 2009.
- Emmerton, C. A., Taube, N., Laceby, J. P., Beundia-Fores, C., Orwin, J., Willis, J.,

```
Jackson, B., Ware, C., Murray, E., Pickering, R., and Ozipko, R.: The WaterSHED monitoring program: (Water: Saskatchewan Headwaters Edmonton and Downstream) technical progress report 2018-2021, Edmonton, 50 pp., 2023.
```

- EPCOR: 2020 Source Water Protection Plan Edmonton's Drinking Water System, 2020.
- Farhaoui, M. and Derraz, M.: Review on Optimization of Drinking Water Treatment Process, J. Water Resour. Prot., 08, 777–786, https://doi.org/10.4236/jwarp.2016.88063, 2016.

904

905

- Fryirs, K.: (Dis)Connectivity in catchment sediment cascades: A fresh look at the sediment delivery problem, Earth Surf. Process. Landforms, 38, 30–46, https://doi.org/10.1002/esp.3242, 2013.
- Giri, S.: Water quality prospective in Twenty First Century: Status of water quality in major river basins, contemporary strategies and impediments: A review, Environ. Pollut., 271, 116332, https://doi.org/10.1016/j.envpol.2020.116332, 2021.
  - Giri, S. and Qiu, Z.: Understanding the relationship of land uses and water quality in Twenty First Century: A review, J. Environ. Manage., 173, 41–48, https://doi.org/10.1016/j.jenvman.2016.02.029, 2016.
- Goode, J. R., Luce, C. H., and Buffington, J. M.: Enhanced sediment delivery in a changing climate in semi-arid mountain basins: Implications for water resource management and aquatic habitat in the northern Rocky Mountains, Geomorphology, 139–140, 1–15, https://doi.org/10.1016/j.geomorph.2011.06.021, 2012.
- Government of Alberta: Alberta River Basins, https://rivers.alberta.ca/, 2023.
- Government of Alberta: Alberta Climate Information System (ACIS),https://acis.alberta.ca/weather-data-viewer.jsp, 2024a.
- Government of Alberta: Water Quality Portal, 903 https://environment.extranet.gov.ab.ca/apps/WaterQuality/dataportal/, 2024b.
  - Government of Canada: Archived hydrometric data, https://www.canada.ca/en/environment-climate-change/services/wateroverview/quantity/monitoring/survey.html, 2023.
- Government of Canada: Canadian Climate Normals 1981-2010,
   https://climate.weather.gc.ca/climate\_normals/index\_e.html, 2024.
- Gulev, S. K., Thorne, P. W., Ahn, J., Dentener, F. J., Domingues, C. M., Gerland, S.,
  Gong, D., Kaufman, D. S., Nnamchi, H. C., Quaas, J., Rivera, J. A.,
  Sathyendranath, S., Smith, S. L., Trewin, B., von Schuckmann, K., and Vose, R.
- Sathyendranath, S., Smith, S. L., Trewin, B., von Schuckmann, K., and Vose, R. S.: Changing State of the Climate System, in: Climate Change 2021: The Physical
- Science Basis. Contribution of Working Group I to the Sixth Assessment Report of
- the Intergovernmental Panel on Climate Change [Masson-Delmotte, V., P. Zhai, A.
- Pirani, S.L. Connors, C. Péan, S. Berger, N. Caud, Y. Che, edited by: Masson-916 Delmotte, V., Zhai, P., Pirani, A., Connors, S. L., Péan, C., Berger, S., Caud, N.,
- Chen, Y., Goldfarb, L., Gomis, M. I., Huang, M., Leitzell, K., Lonnoy, E., Matthews,
- J. B. R., Maycock, T. K., Waterfield, T., Yelekçi, O., Yu, R., and Zhou, B. (eds.)]., Cambridge University Press, Cambridge, UK.,
- https://doi.org/10.1017/9781009157896.004., 2021.
- Hanes, C. C., Wang, X., Jain, P., Parisien, M. A., Little, J. M., and Flannigan, M. D.:
- Fire-regime changes in canada over the last half century, Can. J. For. Res., 49, 256–269, https://doi.org/10.1139/cjfr-2018-0293, 2019.
- 250–269, https://doi.org/10.1139/cjii-2018-0293, 2019.
- Hirsh-Pearson, K., Johnson, C. J., Schuster, R., Wheate, R. D., and Venter, O.:

- Canada's human footprint reveals large intact areas juxtaposed against areas under immense anthropogenic pressure, Facets, 7, 398–419, https://doi.org/10.1139/facets-2021-0063, 2022.
- Hoffmann, U., Hoffmann, T., Johnson, E. A., and Kuhn, N. J.: Assessment of variability and uncertainty of soil organic carbon in a mountainous boreal forest (Canadian Rocky Mountains, Alberta), Catena, 113, 107–121, https://doi.org/10.1016/j.catena.2013.09.009, 2014.
- Kamrath, B., Yuan, Y., Manning, N., and Johnson, L.: Influence of sampling frequency
  and estimation method on phosphorus load uncertainty in the Western Lake Erie
  Basin, Ohio, USA, J. Hydrol., 617, 128906,
  https://doi.org/10.1016/j.jhydrol.2022.128906, 2023.
- Knief, U. and Forstmeier, W.: Violating the normality assumption may be the lesser of two evils, 2576–2590, 2021.

957

- Knox, A. K., Dahlgren, R. A., Tate, K. W., and Atwill, E. R.: Efficacy of Natural Wetlands
  to Retain Nutrient, Sediment and Microbial Pollutants, J. Environ. Qual., 37, 1837–
  1846, https://doi.org/10.2134/jeq2007.0067, 2008.
  - Laceby, J. P., Chung, C., Kruk, M. K., and Kerr, J. .: Evaluation of quality control data from Alberta's lotic water monitoring programs: 2016-2019, Government of Alberta, 71 pp. pp., 2022.
  - Lintern, A., Webb, J. A., Ryu, D., Liu, S., Bende-Michl, U., Waters, D., Leahy, P., Wilson, P., and Western, A. W.: Key factors influencing differences in stream water quality across space, Wiley Interdiscip. Rev. Water, 5, 1–31, https://doi.org/10.1002/WAT2.1260, 2018a.
- Lintern, A., Webb, J. A., Ryu, D., Liu, S., Waters, D., Leahy, P., Bende-Michl, U., and
   Western, A. W.: What Are the Key Catchment Characteristics Affecting Spatial
   Differences in Riverine Water Quality?, Water Resour. Res., 54, 7252–7272,
   https://doi.org/10.1029/2017WR022172, 2018b.
- Loiselle, D., Du, X., Alessi, D. S., Bladon, K. D., and Faramarzi, M.: Projecting impacts
   of wildfire and climate change on streamflow, sediment, and organic carbon yields
   in a forested watershed, J. Hydrol., 590, 125403,
   https://doi.org/10.1016/j.jhydrol.2020.125403, 2020.
  - Lovett, G. M., Burns, D. A., Driscoll, C. T., Jenkins, J. C., Mitchell, M. J., Rustad, L., Shanley, J. B., Likens, G. E., and Haeuber, R.: Who needs environmental monitoring?, Front. Ecol. Environ., 5, 253–260, https://doi.org/10.1890/1540-9295(2007)5[253:WNEM]2.0.CO;2, 2007.
- Lu, X. X., Ran, L. S., Liu, S., Jiang, T., Zhang, S. R., and Wang, J. J.: Sediment loads response to climate change: A preliminary study of eight large Chinese rivers, Int. J. Sediment Res., 28, 1–14, https://doi.org/10.1016/S1001-6279(13)60013-X, 2013.
- Maloney, K. O. and Weller, D. E.: Anthropogenic disturbance and streams: Land use
   and land-use change affect stream ecosystems via multiple pathways, Freshw.
   Biol., 56, 611–626, https://doi.org/10.1111/j.1365-2427.2010.02522.x, 2011.
- Marzeion, B., Cogley, J. G., Richter, K., and Parkes, D.: Glaciers. Attribution of global glacier mass loss to anthropogenic and natural causes., Science, 345, 919–921, https://doi.org/10.1126/science.1254702, 2014.
- Moore, R. D., Fleming, S. W., Menounos, B., Wheate, R., Fountain, A., Stahl, K., Holm, K., and Jakob, M.: Glacier change in western North America: influences on

- hydrology, geomoprhic hazards and water quality, Hydrol. Process., 23, 42–61, 2009.
- Mosley, L. M.: Drought impacts on the water quality of freshwater systems; review and 974 integration, Earth-Science Rev., 140, 203–214, 975 https://doi.org/10.1016/j.earscirev.2014.11.010, 2015.
- Mudryk, L., Santolaria-Otín, M., Krinner, G., Ménégoz, M., Derksen, C., Brutel-Vuilmet,
  C., Brady, M., and Essery, R.: Historical Northern Hemisphere snow cover trends
  and projected changes in the CMIP6 multi-model ensemble, Cryosphere, 14, 2495–
  2514, https://doi.org/10.5194/tc-14-2495-2020, 2020.

- Neary, D. G., Ice, G. G., and Jackson, C. R.: Linkages between forest soils and water quality and quantity, For. Ecol. Manage., 258, 2269–2281, https://doi.org/10.1016/j.foreco.2009.05.027, 2009.
- Newton, B. W. and Taube, N.: Regional variability and changing water distributions 984 drive large-scale water resource availability in Alberta, Canada, Can. Water Resour. 985 J., 48, 300–326, https://doi.org/10.1080/07011784.2023.2186270, 2023.
  - Newton, B. W., Farjad, B., and Orwin, J. F.: Spatial and temporal shifts in historic and future temperature and precipitation patterns related to snow accumulation and melt regimes in Alberta, Canada, Water (Switzerland), 13, https://doi.org/10.3390/w13081013, 2021.
  - Orwin, J. F. and Smart, C. C.: Short-term spatial and temporal patterns of suspended sediment transfer in proglacial channels, Small River Glacier, Canada, Hydrol. Process., 18, 1521–1542, https://doi.org/10.1002/hyp.1402, 2004.
  - Orwin, J. F., Lamoureux, S. F., Warburton, J., and Beylich, A.: A framework for characterizing fluvial sediment fluxes from source to sink in cold environments, Geogr. Ann. Ser. A Phys. Geogr., 92, 155–176, https://doi.org/10.1111/j.1468-0459.2010.00387.x, 2010.
  - Orwin, J. F., Klotz, F., Taube, N., Kerr, J. G., and Laceby, J. P.: Linking catchment structural units (CSUs) with water quality: Implications for ambient monitoring network design and data interpretation, J. Environ. Manage., 312, 114881, https://doi.org/10.1016/j.jenvman.2022.114881, 2022.
  - Palmer, M. A., Lettenmaier, D. P., Poff, N. L., Postel, S. L., Richter, B., and Warner, R.: Climate change and river ecosystems: Protection and adaptation options, Environ. Manage., 44, 1053–1068, https://doi.org/10.1007/s00267-009-9329-1, 2009.
- Pomeroy, J. W., Stewart, R. E., and Whitfield, P. H.: The 2013 flood event in the South Saskatchewan and Elk River basins: Causes, assessment and damages, Can. Water Resour. J., 41, 105–117, https://doi.org/10.1080/07011784.2015.1089190, 2016.
- Price, J. I. and Heberling, M. T.: The Effects of Source Water Quality on Drinking Water Treatment Costs: A Review and Synthesis of Empirical Literature, Ecol. Econ., 151, 195–209, https://doi.org/10.1016/j.ecolecon.2018.04.014, 2018.
- Raymond, P. A., Saiers, J. E., and Sobczak, W. V.: Hydrological and biogeochemical controls on watershed dissolved organic matter transport: Pulse- shunt concept, Ecology, 97, 5–16, https://doi.org/10.1890/14-1684.1, 2016.
- Rice, E.W. Baird, R.B., Eaton, A. D.: Standard Methods for the Examination of Water and Wastewater, 23rd Edition, American Public Health Association, American Water Works Association, Water Environment Federation, Washington, D.C., 2017.

- Rood, S. B., Pan, J., Gill, K. M., Franks, C. G., Samuelson, G. M., and Shepherd, A.:
  Declining summer flows of Rocky Mountain rivers: Changing seasonal hydrology
  and probable impacts on floodplain forests, J. Hydrol., 349, 397–410,
  https://doi.org/10.1016/j.jhydrol.2007.11.012, 2008.
- Runkel, R. L., Crawford, C. G., and Cohn, T. a: Load Estimator (LOADEST): A
   FORTRAN program for estimating constituent loads in streams and rivers, Tech.
   Methods. U.S. Geol. Surv. U.S. Dep. Inter., 2004.
- Shi, P., Zhang, Y., Li, Z., Li, P., and Xu, G.: Influence of land use and land cover patterns on seasonal water quality at multi-spatial scales, Catena, 151, 182–190, https://doi.org/10.1016/j.catena.2016.12.017, 2017.
- Shi, X., Mao, D., Song, K., Xiang, H., Li, S., and Wang, Z.: Effects of landscape changes on water quality: A global meta-analysis, Water Res., 260, 121946, https://doi.org/10.1016/j.watres.2024.121946, 2024.
- Sidak, Z.: Rectangular Confidence Regions for the Means of Multivariate Normal Distributions, 62, 626–633, https://doi.org/10.1080/01621459.1967.10482935, 1967.
- Simpkins, G.: Snapshot: Extreme Arctic heat, Nat. Clim. Chang., 7, 95, https://doi.org/10.1038/nclimate3213, 2017.
- Sinha, R. and Friend, P. F.: River systems and their sediment flux, Indo-Gangetic plains, Northern Bihar, India, Sedimentology, 41, 825–845, https://doi.org/10.1111/j.1365-3091.1994.tb01426.x, 1994.
- La Sorte, F. A., Johnston, A., and Ault, T. R.: Global trends in the frequency and duration of temperature extremes, Clim. Change, 166, 1–14, https://doi.org/10.1007/s10584-021-03094-0, 2021.
  - Spencer, R. G. M., Butler, K. D., and Aiken, G. R.: Dissolved organic carbon and chromophoric dissolved organic matter properties of rivers in the USA, J. Geophys. Res. Biogeosciences, 117, https://doi.org/10.1029/2011JG001928, 2012.
- St-Hilaire, A., Duchesne, S., and Rousseau, A. N.: Floods and water quality in Canada: A review of the interactions with urbanization, agriculture and forestry, Can. Water Resour. J., 41, 273–287, https://doi.org/10.1080/07011784.2015.1010181, 2016.
- Staniszewska, K. J., Cooke, C. A., and Reyes, A. V.: Quantifying meltwater sources and contaminant fluxes from the Athabasca Glacier, Canada, ACS Earth Sp. Chem., 5, 23–32, https://doi.org/10.1021/acsearthspacechem.0c00256, 2021.
  - Statistics Canada: Census Profile. 2021 Census of Population, Ottawa, https://doi.org/98-316-X2021001, 2023.
- Su, H., Cheng, L., Wu, Y., Qin, S., Liu, P., Zhang, Q., Cheng, S., and Li, Y.: Extreme storm events shift DOC export from transport-limited to source-limited in a typical flash flood catchment, J. Hydrol., 620, https://doi.org/10.1016/j.jhydrol.2023.129377, 2023.
- Tetzlaff, D., Laudon, H., Luo, S., and Soulsby, C.: Ecohydrological resilience and the landscape water storage continuum in droughts, Nat. Water, 2, 915–918, https://doi.org/10.1038/s44221-024-00300-y, 2024.
- Townsend-Small, A., McClain, M. E., Hall, B., Noguera, J. L., Llerena, C. A., and Brandes, J. A.: Suspended sediments and organic matter in mountain headwaters of the Amazon River: Results from a 1-year time series study in the central Peruvian Andes, Geochim. Cosmochim. Acta, 72, 732–740,
- https://doi.org/10.1016/j.gca.2007.11.020, 2008.

1041

1042

1049

- USEPA: Clean Water Act Section 303(d): Impaired Waters and Total Maximum Daily1064 Loads (TMDLs), 2024.
- Veldkamp, E., Schmidt, M., Powers, J. S., and Corre, M. D.: Deforestation and reforestation impacts on soils in the tropics, Nat. Rev. Earth Environ., 1, 590–605, https://doi.org/10.1038/s43017-020-0091-5, 2020.
- Vercruysse, K., Grabowski, R. C., and Rickson, R. J.: Suspended sediment transport dynamics in rivers: Multi-scale drivers of temporal variation, Earth-Science Rev., 1070 166, 38–52, https://doi.org/10.1016/j.earscirev.2016.12.016, 2017.
- Vickers, G., Schmidt, D., and Mullock, J.: The weather of the Canadian prairies. Graphic Area Forecast 32, 228 pp., 2001.
- van Vliet, M. T. H., Thorslund, J., Strokal, M., Hofstra, N., Flörke, M., Ehalt Macedo, H., Nkwasa, A., Tang, T., Kaushal, S. S., Kumar, R., van Griensven, A., Bouwman, L., and Mosley, L. M.: Global river water quality under climate change and hydroclimatic extremes, Nat. Rev. Earth Environ., 4, 687–702, https://doi.org/10.1038/s43017-023-00472-3, 2023.
- Vogel, M. M., Zscheischler, J., Wartenburger, R., Dee, D., and Seneviratne, S. I.:
   Concurrent 2018 Hot Extremes Across Northern Hemisphere Due to Human Induced Climate Change, Earth's Futur., 7, 692–703,
   https://doi.org/10.1029/2019EF001189, 2019.
- Walsh, J. E., Ballinger, T. J., Euskirchen, E. S., Hanna, E., Mård, J., Overland, J. E.,
  Tangen, H., and Vihma, T.: Extreme weather and climate events in northern areas:
  A review, Earth-Science Rev., 209, 103324,
  https://doi.org/10.1016/j.earscirev.2020.103324, 2020.
- Ward, N. D., Bianchi, T. S., Medeiros, P. M., Seidel, M., Richey, J. E., Keil, R. G., and Sawakuchi, H. O.: Where carbon goes when water flows: Carbon cycling across the aquatic continuum, Front. Mar. Sci., 4, 1–28, https://doi.org/10.3389/fmars.2017.00007, 2017.
- Whitehead, P. G., Wilby, R. L., Battarbee, R. W., Kernan, M., and Wade, A. J.: A review
  of the potential impacts of climate change on surface water quality, Hydrol. Sci. J.,
  54, 101–123, https://doi.org/10.1623/hysj.54.1.101, 2009.
- Wood, P. J. and Armitage, P. D.: Biological effects of fine sediment in the lotic
  environment, Environ. Manage., 21, 203–217,
  https://doi.org/10.1007/s002679900019, 1997.

- Yin, J., Gentine, P., Zhou, S., Sullivan, S. C., Wang, R., Zhang, Y., and Guo, S.: Large increase in global storm runoff extremes driven by climate and anthropogenic changes, Nat. Commun., 9, https://doi.org/10.1038/s41467-018-06765-2, 2018.
- Yin, S., Gao, G., Ran, L., Lu, X., and Fu, B.: Spatiotemporal Variations of Sediment
  Discharge and In-Reach Sediment Budget in the Yellow River From the Headwater
  to the Delta, Water Resour. Res., 57, 1–24, https://doi.org/10.1029/2021WR030130,
  2021.