# Peer review of "Hydrometeorology and landscapes control sediment and dissolved organic carbon mobility across a diverse and changing glacier-sourced river basin"

_EGUsphere, 2025_

## Author Comment (AC1)

**Hydrometeorology and landscapes control sediment and organic matter mobility across a diverse and changing glacier-sourced river basin**

*Craig A. Emmerton[1]\*, John F. Orwin[2], Cristina Buendia[1α], Michael Christensen[3], Jennifer A. Graydon[1], Brian Jackson[4], Elynne Murray[4], Stephanie Neufeld[3], Brandi W. Newton[2], Ryan Ozipko[4], Rick Pickering[4], Nadine Taube[2], Chris Ware[4]*

**Response to Referees Document**

**Author comments:** We sincerely thank both anonymous reviewers for taking the time to review and provide helpful suggestions to improve our manuscript. In this response document, the original comment will be in bold-type and the response will be below each comment in red normal type.
* * *
**RC1: Anonymous Referee #1, 18 Jul 2025** https://doi.org/10.5194/egusphere-2025-1971-RC1

**This study addresses the impact of hydrometeoerological extremes on various landscapes – particularly with respect to TSS and DOC. Uses North Saskatchewan River between 2019-2022 as case study. Study finds that warming and wetter periods resulted in greater TSS and DOC export, with DOC behaviour being more predictable than TSS during these periods.**

**I could easily understand the paper and it was the methods seem sound and appropriate for answering the key research questions of the impact of hydrometerological extremes on TSS and DOC export across different landscapes.**
Thank you.

**I only have a few suggestions that will hopefully make it easier for the reader to fully appreciate and understand the study and its findings.**

**Line 41-42: "During higher water" -> consider changing to "During wetter years"**
This will be changed as suggested.

**Line 53: "Recently" -> consider specifying when**
We will add "since the 1980s" in place of "recently".

**Line 103: "Ongoing changes" -> for international audiences, please state what changes are being experienced**
We will change this to, "In addition to ongoing changes in summer flows and sediment yields experienced by Rocky Mountain rivers…".

**Line 105: "recently endured" -> for international audiences, please specify when these extremes have been occurring (how recent have they been?)**
Thank you for the comment. This line is a bit of a teaser for the rest of the paper, so it's these extremes that form the basis of the paper. We want to leave description of the extremes to other portions of the paper. However, we will add "2019-2022" in parentheses following "recently" in that sentence to make the connection with the rest of the paper.

**Line 105: "High runoff years" -> please specify how high the runoff is**
Please see previous response. We go through these specifics through the rest of the paper, so we feel that information is best left off this sentence.

**Line 245: "yet" -> implies that one day these data will be available.**
Correct. These data, in fact, are now available since the submission of this paper. However, considering that this paper does rely on historic data, these sites are still not considered for this paper since they only have 4 years of record accumulated.

**Line 252: "Phosphorus" -> in this sentence, unclear if it is total and/or dissolved phosphorus here**
Thank you for your comment. This will be revised to, "…while DOC concentrations showed patterns similar to total and dissolved forms of nitrogen and phosphorus, and…"

**Line 285-287: "The three fixed factors…autoregressive approach" -> Having the equation structure shown here might help clarify how the model was set up for the reader.**
Thanks for the helpful comment. The equation will be added in-text with some minor changes to surrounding words to help explain the parts of the equation.

**Line 288: "normality standards" -> would like some indication of whether they actually ended up meeting normality standards, or whether they just got a little bit closer to normality after transformation**
For our TSS and DOC yield datasets, our data were strongly and positively skewed and failed multiple normality tests with high values for skewness and kurtosis. Log transformation of these values resulted in much more defined bell-shaped curves, with improved skewness, but still rejection of normality statistics in several cases. Similar findings were observed for the dissolved organic matter model (skewness, etc.). We were happy with these transformations considering that linear mixed models are inherently robust to slight to moderate non-normality and do not typically impose interpretation problems for linear mixed models (Knief and Forstmeier, 2021). We will update this sentence to, "Daily catchment yield data (dependent variable) were strongly and positively skewed and were log-transformed to improve normality to meet linear mixed model standards (Knief and Forstmeier, 2021).".

**Line 289: "Sidak multiple comparisons" -> please provide a citation for this as I (and other readers) might not be familiar with this approach.**
We will add a reference to the original paper outlining this multiple comparison approach (Sidak, 1967).

**Line 295: This is a great way to get around the fact that many catchment characteristics are cross-correlated with each other**
Thank you for the kind comment.

**Line 337: "NSR-Edmonton Station" -> is this the downstream most station? Would be good to specify where this is in the context of the catchment for readers unfamiliar with the area.**
Good observation. No, this is not the most downstream flow station. We will add reference to Figure 1 here, which shows the location of the NSR-Edmonton flow station.

**Line 352: "substantially" -> would be good to quantify this here**
Thanks for the comment. We will add, "(99th percentile of flows)" after "substantial".

**Line 383: "statistically" -> I'm interpreting this as the slope having p<0.05**
We should be more explicit here. We are linking to Table S3 that shows both the coefficient of determination for the model (r2; assesses the variability in the dependent variable explained by the independent variable) and the statistical significance of the F-statistic in the model. For clarity, we will remove, "and statistically" and will improve the caption in Table S3 to:

"**Table S3** Loadest and log-log regression export model concentration-flow relationships for total suspended solids (TSS) and dissolved organic carbon (DOC) from the North Saskatchewan River (NSR) mainstem (emphasized) and its tributaries from 2019 to 2022 water years. $r^2$ is the coefficient of determination of the regression and Loadest models and statistical significance ($\alpha$=0.05) of the model, as determined by the F-statistic, is denoted by an asterisk. C-cordillera; F-foothills; MA-mixed agriculture; A-agriculture."

**Line 387: "statistically stronger" -> I'm not actually sure what this means – does this mean that the slopes are steeper?**
Thanks for this helpful comment. Similar to the previous comment, we have removed "statistically" here and since this sentence still refers to Table S3, we will rely on the new Table S3 caption to properly explain statistical significance.

**Figure 4: I'm struggling to understand what the letters mean in the mixed modelling results here. If you can provide an explanation in the caption, it would be helpful.**
Thanks for your comment and it is a fair critique. We will update the caption as follows to help out the reader, "Linear mixed model results (right of graphs) assess statistical differences between landscapes, water years, and their combination across all export models. Different letters within a given category (i.e., landscape, water year) denote statistical differences as determined by the linear mixed model."

**Line 524: "C-Q" -> these relationships feature a lot throughout especially the discussion. I wonder whether it is worth having a figure/table of these relationships in the main text instead of just in the supplementary materials. Or at least putting some examples of the relationships in the text.**
Thanks for the comment. We will include a new supplementary figure that shows the C-Q associations (as biplots) for all the stations to support the reader.

**Line 545: "we observed…changes in flow" -> at this point would like to see which specific result this finding is being drawn from. I think this is pretty important, but am unclear as to how the results of the analysis have led to this conclusion.**
Thanks for this comment. Though you can dig through our export tables in the supporting information or compare differences between years in figures 4 and 5, we rely on Table S7 to specifically show the threshold-type behaviour observed in some rivers between years relative to the amount of flow occurring in the rivers. Your point is taken that this is quite important to the manuscript, so we suggest moving this table to the main manuscript.

**Line 552: "model results" -> I assume this refers to the linear mixed models, but would be good to specify**
Thank you and good comment. We will add "export" in front of "model" in this sentence to link with the four models used to determine loading, which were outlined in the Methods section.

**Line 553: "high end" -> I assume this means high C and Q?**
Yes, that is correct. We will alter the sentence to make that more explicit. The sentence will be updated to, "…events perturb model fits at high flows and concentrations of C-Q associations."

**Line 583: "weaker" -> again here - providing some numbers from comparison (as evidence) would be really useful to help the readers understand the conclusions being made.**
We will add reference to Table S3 in this sentence to remind the reader where to find the numbers related to this comparison statement.

**Line 592: "stronger C-Q associations" -> I assume this means a steeper C-Q slope?**
This actually means that the coefficient of determination is larger, which describes more variability is captured by the fitting of the model to the data. To be more explicit, we will update this line to, "…cordillera rivers had stronger C-Q associations (i.e., larger coefficients of determination; Table S3) but did not produce…".

**Line 621-623: "little evidence in…DOC mobility" -> would be great to refer specifically back to what findings/results points to this not happening**
Sure, we can refer back to Figure 6 and the off-channel "unmonitored" additions.

**Table S4-S5: would be really good to show error bars in load estimations too (max-min)**
Thanks for the comment. We will attempt to fit these in this table, but space requirements may dictate that we split this table to show export and yield separately.

---

## Author Comment (AC2)

**Hydrometeorology and landscapes control sediment and organic matter mobility across a diverse and changing glacier-sourced river basin**

Craig A. Emmerton[1]*, John F. Orwin[2], Cristina Buendia[1α], Michael Christensen[3], Jennifer A. Graydon[1], Brian Jackson[4], Elynne Murray[4], Stephanie Neufeld[3], Brandi W. Newton[2], Ryan Ozipko[4], Rick Pickering[4], Nadine Taube[2], Chris Ware[4]

**Response to Referees Document**

**Author comments:** We sincerely thank both anonymous reviewers for taking the time to review and provide helpful suggestions to improve our manuscript. In this response document, the original comment will be in bold-type and the response will be below each comment in red normal type.
* * *
RC2: Anonymous Referee #2, 28 Jul 2025 https://doi.org/10.5194/egusphere-2025-1971-RC2

**The authors assess how hydrometeorological conditions and catchment structure influence total concentration and fluxes of total suspended sediment (TSS) and dissolved organic carbon (DOC) in a very large glacial catchment (upper North Saskatchewan River). I enjoyed reading the manuscript and I think it falls within the scope of HESS. It makes an important data contribution, and the authors made a big effort to summarise and nicely show a large amount of data and results. The paper is well written. However, I have a few concerns that I would have liked the authors to better addressed. The most relevant one related to the method and way they chose to predict and report annual fluxes.**
Thank you.

**I find the title misleading, "organic matter mobility" might also include particulate forms, whereas the article only looks at dissolved organic carbon. There are also other forms than carbon contributing to dissolved organic matter. Consider replacing by "dissolved organic carbon" to be precise. This should also be revised in the manuscript.**
Thank you for the comment. The title was originally stated this way as the vast majority of total organic carbon measured in the rivers of the NSR system is in the dissolved form. However, to be more explicit, we will update the title to, "Hydrometeorology and landscapes control sediment and dissolved organic carbon mobility across a diverse and changing glacier-sourced river basin". We will also replace "dissolved organic matter" with "dissolved organic carbon" throughout the manuscript.

**The paper is missing a clear research question and/or hypothesis. The authors "assess how fluctuations in hydrometeorological conditions (…) impact the downstream delivery" of DOC and sediments. However, I think there is not enough information in the introduction explaining what is already known about the processes and leading mechanisms, and which knowledge gaps the research intends to fill. The authors expect "differences between TSS and DOC mobilized through the river network (…)" what remains very vague and obvious. This could be rephrased into a more precise hypothesis, and I encourage the authors to formulate a research question.**
Thank you for this comment and will help in clarifying some of our language. Our hypothesis was perhaps too concealed at the end of the Introduction (i.e., We expected differences between…on

line 115). Also, this hypothesis does not do a great job at separating out the two assessments in the paper: one of different catchment yields from various landscapes and years, and one of mass export through the entire river network between years. Thus, we will modify the last paragraph of the Introduction to have clearer research questions and hypotheses. Also, we appreciate the comment about the lack of information known about what is already known about these processes. Though we feel river water quality responses to changes in runoff across landscapes is well established in the literature, we will add a line referencing a recent comprehensive review paper on climate change and land-use impacts on river water quality that calls for more studies that disentangle climate from landscape impacts on water quality.

Thus, we will modify the last sentence of the second-to-last paragraph of the Introduction to:

"Monitoring programs that include these features are rare but present an opportunity to better understand how climate variability and change across diverse landscapes can impact major rivers and water users; a key gap in our understanding of river water quality (van Vliet et al., 2023)."

We will also modify the last paragraph of the Introduction to:

"Here, we concurrently assess how widely varying hydrometeorological and landscape conditions of a diverse and changing northern river basin impact the downstream delivery of key water quality, aquatic ecosystem health, and drinking water treatment parameters (i.e., TSS, DOC). The North Saskatchewan River (NSR) in western Canada is sourced from the glaciated Canadian Rocky Mountains and transects mountain bedrock, forest, agricultural, and urbanized landscapes. In addition to ongoing changes in summer flows and sediment yields experienced by Rocky Mountain rivers (Goode et al., 2012; Rood et al., 2008), the NSR basin has recently (i.e., 2019-2022) endured hydrometeorological extremes, ranging from high runoff years to historic heat and drought conditions. These wide-ranging conditions may become more common in the future and thus provide a window to better understand the response of northern rivers to climate extremes. We used water quality and quantity data from a nested monitoring program for this study. The WaterSHED (Water: Saskatchewan Headwaters Edmonton Downstream) Monitoring Program is a collaborative tributary catchment monitoring network across the NSR basin initiated in 2019 by a drinking water treatment utility, local governments, and a non-government organization (Emmerton et al., 2023). The program was designed to balance monitoring across different catchment landscape conditions within the NSR basin. In addition, long-term water quality and flow monitoring data were used from stations on the downstream NSR mainstem, operated by the governments of Alberta and Canada. We had two goals of this study: 1. to distinguish the impacts of interannual hydrometeorological variability from changes in landscape conditions on catchment yields of sediment and dissolved organic carbon; and 2. to determine the relative roles of catchment and in-channel contributions of a major river (i.e., NSR) on interannual export of sediment and dissolved organic carbon from the basin. We hypothesized that TSS (i.e., particulates) and DOC (i.e., organic material) yields would show similar interannual responses to changes in hydrometeorology across the network, but differences between landscape types. We also hypothesized that the degree of influence of in-channel sources and sinks of sediment and DOC relative to catchment inputs would change between wetter and drier years."

We will also address the outcomes of the two clearer hypothesis statements in the appropriate section of the Discussion or Conclusion.

**Until I got to 2.3.1 I thought the definition of CSUs was "to quantify the influence of major landscape types on modulating hydrometeorological conditions" (lines 142-143). However, it is then presented in section 2.3.1 that the method is used to classify (with replicates) the catchments in different types ("the number of clusters selected for further assessment was chosen manually to balance replication of catchments within a reduced number of clusters"), is this right? I think the aim of the analysis should be better presented earlier (in section 2.2.1). I also find a bit confusing that sections 2.2.1 (data) and 2.3.1 (numerical assessments) are separated. I also was confused in Figure 1. Maybe a cross-reference to 2.3.1 could be added in the figure caption.**

Thank you, these are important observations. We wanted to separate out data sources (2.2) from data assessments (2.3) in the Methods. However, the language used at the start of 2.2.1 may not focus enough on the data comprising CSUs.  We will change the leading sentence section 2.2.1 to:

"With a goal to classify different landscape types of monitored catchments across NSR basin (section 2.3), we used geospatial data related to the catchment structural unit (CSU) approach (Orwin et al., 2022). CSUs are a series of unique numerical identifiers derived from geospatial data of land use/landcover, surficial geology, wetlands, and slope that are known to play significant roles in how catchments respond to hydrometeorological inputs (Orwin et al., 2022). CSU codes are generated at a 400 $m^2$ pixel scale across the basin, to provide wall-to-wall coverage. For example, a pixel code of 60111 represents a dominant shrubland cover (60,000), mixed coarse-fine surface geology (100), with wetlands present (10), and a slope less than 10% (1). Further data details are provided in the supporting information and Table S1a."

We will modify the Figure 1 caption to better link CSUs to the goal of classifying landscape conditions across the NSR basin and a reference to Methods section 2.2.3:

"**Figure 1** The upper North Saskatchewan River (NSR) basin in Alberta, Canada showing tributary catchment and NSR mainstem water quality and flow stations. Catchment structural units (CSUs; see Methods section 2.2.1) showing general landscape cover and land use across the basin are shown. Landscape conditions of monitored catchments are classified using hierarchical clustering analysis (lower inset; see Methods section 2.3.1). Greyed catchment names contributed to the landscape clustering analysis but were not included in the broader study due to insufficient flow data. Of note, though NSR Whirlpool Point is on the NSR mainstem, it is viewed in this study as a cordillera tributary as the river's origin is part of this catchment. Additional data from Shaded Relief (http://www.shadedrelief.com/world_relief/home.html)."

**Figure S1 shows the sampled flow percentiles. Which flow data did you use to make this figure (15-min, hourly, daily, monthly?). Without this information it is very difficult to interpret the significance of the figure. It seems impressive to be able to sample 90%s of historic flow values in most stations with 4-years monthly samples. Maybe this is achieved with the "flow-targeted sampling". I think more information is needed.**

We used the daily flow on the day of each water quality collection event on each river. Figure S1 is meant to convey the range of percentile flows of the historic record experienced on the river while

collecting samples. So there are ~30 sampling points distributed across each bar, with the minimum and maximum sampled flows occurring that the bounds of each bar. We felt it was important to show that our sampling just didn't capture "regular" conditions in the rivers. We will update the caption to, "Minimum and maximum daily flow percentiles of the historic record experienced during collection of all water samples between 2019 and 2022 water years."

**I think the methods used to predict annual SSC and DOC loads should be better explained. So far there is only reference to the R package loadflex (Appling et al 2015 – a reference that does not appear in the reference list). Results are expressed as "Export ± standard error" (Tables S9 and S10), the reader should know how this is computed. With the given information it is very difficult to address the difference between methods and their accuracy and precision.**
Thank you for finding the missing reference to Appling et al. 2015. This entry will be added to the reference list. Originally, we leaned on the Appling reference to explain the specific details of the LoadFlex tool, and we supplemented this with more details in the Supporting Information. However, we agree to add more information here in the main text, either from the SI section, or from the Appling paper (e.g., uncertainty). The new part of the paragraph will read as follows:

"…..using several approaches. We used the R package *loadflex* (Appling et al., 2015) to estimate annual export at catchment and NSR mainstem monitoring stations using four different export models. The first model uses a rectangular interpolation approach that connects consecutive concentration measurements using a period-weighted average approach. Daily export is quantified using the product of filled concentrations and flow data on a given day. The second model is a log regression model between periodically measured concentrations and its paired flow data on the day of collection. This model is then used with the full flow data record to quantify daily export estimates. The third model is the United States Geological Survey Loadest model (Runkel et al., 2004).  This approach fits several different regressions to periodic concentration data and paired flow to find the best performing regression model to fit the data. This model is then used with flow data to quantify daily export. Finally, the composite model uses a regression model (Loadest as default) and collected concentration data to generate a residual dataset between regression estimates and real data. These residuals are then used systematically to correct regression estimates to provide an improved regression that is used with flow to quantify daily export. Daily uncertainty is quantified as a standard error of prediction (SEP) for each model as outlined in Appling et al. (2015). Briefly, interpolation models use a jackknife process where interpolation models are fit to the data not including one observation and differences between the removed data point and the predicted point are quantified. For regression-based models (i.e., regression, Loadest), daily SEP is determined as the square root of the sum of coefficient uncertainty. The composite model uses a composite parametric bootstrap approach. To produce annual water year export for each site and each model, we summed all daily export estimates for each water year and associated daily SEP via summing propagation of error rules. For errors related to unmonitored flow, we used the mean proportional error of the river(s) used to calculate unmonitored flows via their yields (see Methods). Errors related to the change in storage ($\Delta S$) term were determined by simple error propagation rules. Annual tributary catchment yields (mass per areas) were quantified…."

As a result of taking information from the SI section "Mass Export Models" and considering a comment below, we will include a new uncertainty section in the SI to talk about uncertainty of using *loadflex* in this study (see comment below).

**Loadest and log-log regression export model concentration-flow relationships fits for the 2019-2022 periods are reported in Table S3, but what about Loadset composit? Also, there is a few manuscripts investigating the uncertainties associated with these methods, which are rather high, so this should be better addressed. Annual fluxes (and storage) are calculated/reported using the median export of the four models (supplementary tables S4 and S5). I think this adds an extra layer of bias in the results that could be avoided by finding out which method works best (an analysis that is completely missing). I do not think using the median of the four methods included in the loadflex R package is the most accurate way to report annual fluxes of DOC and TSS. This is especially true because r2 values for some sites are highly variable when using different methods, e.g. 0.17 - 0.99 (DOC Strawberry Ck.). The authors also provide the values reported with the four methods, but it is very difficult to assess how good the methos are when only looking at the results. The same happens when looking at the supplementary material, where only r2 values and total fluxes (also per year) are reported. Maybe figures would help here.**

Thank you for the comment. The goal of Table S3 was to show the reader the strength of simple regressions between flow and concentration to assess strength in these relationships (this will now be supported by new C-Q graphs in the SI). Loadest and regression models are both regression-type assessments with typical linear-fit statistics such as $r^2$. The interpolation and composite assessments have no such statistics. Because this table was not meant to assess differences between the models, rather give the reader grounding in the simple flow versus concentration relationships, we did not add other statistics to this table.

Uncertainty both within models (at a given site) and between models is an important issue that we can address better, particularly in the SI section. In terms of how uncertainty is calculated *within* each model, we feel the Appling et al. (2015) paper and the code do a nice job outlining how that is assessed. In our export tables S9 and S10, we see a mix of relatively low uncertainties around calculated export, and some higher uncertainties, particularly if the rivers have the potential to transport lots of suspended sediment in a given year or if a river transports little sediment and C-Q relationships are weaker. However, differences between years at a given site, between rivers of a given landscape type, and between rivers between landscape types are often much larger than these errors. Also, we see the same patterns across the NSR basin regardless of which model we use, so we are confident our assessments are finding real differences between years and landscape types, despite some higher uncertainties around our numbers in Table S9 and S10. This is the main reason we did not specifically assess the strength of model performance in *loadflex*, as other papers, as you mention, discuss this (e.g., Kamrath et al., 2023). We will attempt to improve the uncertainty section in the SI to better discuss uncertainty related to our use of *loadflex* (see in comment below).

The use of the median of all models (our calculation, not from *loadflex* specifically) was strictly for the export portion of the paper (i.e., not the linear mixed model), which integrates tributary inputs, unmonitored inputs, and the mainstem NSR stations and was not a statistical exercise assessing different models. This exercise was to show how the mainstem of the NSR was influencing what ultimately was being carried by the major river versus the tributaries. Yes, we could have picked a specific model, but in terms of absolute export, there were few changes in the patterns of export throughout the network between models, so we felt using the median of all models was a conservative approach.

**Figure would also help when the authors discuss how extreme high events perturb model fits at the high end of C-Q associations (lines 551-553). This is very difficult to assess as we do not see the data at all. Maybe an hydrograph with the sampling points would also help to evaluate how representative are the sampling times.**

Thanks, and this is a great suggestion. We will add an SI figure showing the C-Q relationships for TSS and DOC for each of the rivers we have sampled.

**Is there any other evidence of in-channel/bed or bank erosion or deposition that could be used to cross-check the results? Sediment storage in the upper two station reaches seem to be happening independently of runoff (lines 484-487), is this related to the dam?**

Thanks, and good question. This is a simple In/Out/Storage mass balance model across a large river network. Since we quantified the In (tributaries; monitored and unmonitored) and Out (NSR mainstem) well, the remainder must be from stores in the NSR mainstem or its banks and marginal riparian areas, or from groundwater (typically low in TSS and DOC). We do not have any other direct evidence that is not anecdotal on this river system, however, sediment transport processes in glacier-impacted rivers are well known to store and transport sediment in-river (we reference some of these studies in the manuscript). We also see evidence of this as storage in any given reach can typically show positive and negative values. Regarding sediment storage independence of runoff, this can be expected as sediment in rivers often shows a threshold-type behavior as erosion of banks and point bars often need a minimum energy to erode (Table S7). Also, as you suggest, there will be disconnects between flow and sediment transport in the uppermost reach of the NSR where a large dam exists.

**The authors discuss the projection and consequences of their results in the future (lines 653-705). I find some of the discussion going a bit too far. For instance, the authors discus in lines 663-674 how changes in DOC and SSC will impact water treatment cost in the future or ecosystem health. I can see the importance of this, but I think it goes beyond the scope of the paper to describe future needed adjustments in water treatment facilities.**

Thank you for the comment. We have co-authors on this paper from Epcor Water Canada who are well integrated into the daily operations of drinking water treatment plants in the Edmonton area. Daily and seasonal changes in DOM concentrations in the North Saskatchewan River indeed have important cost implications for the production of safe drinking water. For instance, increases in DOM concentrations during normally expected clear water conditions may cause the plant to alter how they treat water with subsequent costs in chemical additions. This is an important consequence of changes in runoff and in-stream concentrations that we feel is worth noting. Much of this work has been driven by expected climate-related changes in the NSR basin and future outlooks for water users, which is why we have provided some high-level guidance.

**I also miss a clearer evaluation of the limitations of the used approach and acknowledge of the uncertainties involved. Especially to address the uncertainties associated with the sampling and the estimation of annual loads. Between 37 and 55 samples (four years) are used to estimate annual loads (between 9 and 14 samples per year). The authors should be clearer about the high uncertainties associated with the results.**

Thank you for this comment. We agree that a section discussing these uncertainties should be included in the SI (as outlined in a comment earlier). We will update the SI section about mass export models as follows:

"**Mass export models and uncertainty**
Quantifying export of suspended and dissolved material of rivers presents challenges and can produce widely different estimate precisions (Aulenbach et al., 2016). The typical sampling strategy to quantify export in rivers is to periodically sample for water quality at the same location where high-frequency flow monitoring is occurring and producing average daily flow rates. Manual water sampling of many rivers across large areas, such is the case in the North Saskatchewan River basin, is a laborious and expensive effort typically leading to monthly or biweekly sampling. This approach means daily or sub-daily changes in flows and water quality, typically during fleeting storm events, can be missed. This leads to less data for models to draw from particularly when flows and water quality concentrations can be high together, with potential impacts on model precision. Aside from fullness of datasets, how you quantify export can be performed differently (e.g., regression, interpolation, etc.) and can also impact uncertainty as well. *Loadflex* uses four different modelling approaches to estimate export from rivers; each having advantages and disadvantages affecting the accuracy and precision of export estimates (Kamrath et al., 2023).

In our study, our typical accumulation of samples for a given river was between 30 and 50 over a four-year period, but with samples from a majority of flow conditions experienced historically in each river (Figure S1) and a sampling bias towards higher flow conditions with enhanced freshet monitoring. From this, we expect improved daily model precisions in *loadflex* but uncertainty around annual export (reported as standard error of prediction) was typically between 10 and 20% for each of our stations (Tables S9, S10). Indeed, there were some TSS or DOC-poor rivers that did not have strong concentration-discharge associations and therefore had much more severe uncertainties. This reduces our confidence in export calculations of individual river stations in any given year. However, we also observed that variability of export measured: 1. between years of a given river; 2. between rivers of a given landscape class; 3. between rivers of different landscape classes; and 4. between export models of a given river was often substantial beyond the uncertainty reported for any given river export value. We also observed the same export patterns between years, rivers, and landscapes regardless of the model used (Tables S9, S10), suggesting that uncertainty at the river station level was not impacting the interannual and inter-river yield and export patterns. Therefore, though there is notable uncertainty around the absolute values of export and yields in our rivers, the larger differences in export and yields driven by interannual changes in hydrometeorology and differences between landscapes appears to override this as demonstrated in Tables S9, S10 and our linear mixed model approach."

**I would also advice to revise the conclusions to mainly address the most relevant findings. I do not think that some of the facts described (e.g. that a broader application of monitoring approaches that capture landscape heterogeneity (…) will lead to better decisions) are direct conclusions of the results shown.**
Thank you for the comment. With additional paragraphs and sections being added to in this manuscript, we will work on streamlining the conclusions to a more targeted recount of the paper as follows:

"**Conclusion**
Using our multi-annual, representative monitoring design for the NSR basin tributaries, we showed that catchment yields of TSS and DOC have significant variability not only between catchments of different structural/landscape conditions, but also between those that are similar. This demonstrated the utility of replication of river monitoring even in landscapes considered to have similar landscape conditions or human impacts. At the same time, interannual variability was a strong driver of TSS and DOC yields at all rivers and landscape types, suggesting hydrometeorology was a stronger influence on TSS and DOC mobilization downstream than landscape condition in the NSR basin. At the basin scale, TSS and DOC yields showed similar patterns between landscapes and years, supporting our hypothesis and further highlighting the importance of hydrometeorology in controlling the conditions of tributaries in this basin.

When integrating a major river in a nested monitoring design, we demonstrate the importance of channels, banks, and riparian areas of major rivers in controlling the amount of TSS and DOC stored in and delivered from a basin. This suggests that landscape conditions may not fully explain the water quality of major rivers, nor fully control the mobilization of terrestrial matter, and demonstrating the importance of monitoring major river systems in a nested manner. In support of our hypothesis, the degree of in-channel contributions to basin export was greater during wet periods than dry and highlights how active major rivers are in the storage and transport of terrestrial material.

The NSR in western Canada supplies water to over 1.5 million Canadians and is experiencing change due to continued loss of glacial meltwater inputs, greater heating and rainfall accumulations, and increasing human populations. Our study suggests that NSR waters are expected to become more turbid and enriched in dissolved organic material under this climate regime and will likely cause challenges for water users (e.g., drinking water utilities), river management strategies, and aquatic biota. However, broader application of monitoring approaches that capture landscape heterogeneity, replicate monitoring of diverse catchment conditions, and include entire river basins in an integrated fashion will provide improved data and understanding of river systems and ultimately better decisions when managing water resources."

**MINOR COMMENTS**
**Line 23: In the introduction (lies 83-84) the authors stated the opposite, that there are many factors that challenge the use of rivers as indicators of change.**
Thank you for the comment, that was helpful. Changes of rivers are obviously valuable indicators of climate change, but our point later in the introduction states they are only valuable if monitored well. We will revise the short summary to, "Rivers are valuable indicators of climate change when extensively monitored."

**Line 24: "river material" is too vague.**
This will be changed to, "…understand how suspended sediment and dissolved organic carbon change across differing…".

**Lin 62, 64: I would also recommend using another word as "material" is vague.**
It is challenging to include all the different types of terrestrial material that could be directed to rivers (e.g., organic and humic soil material, litterfall, many forms of organic matter, inorganic soil

particles, nutrients, etc.). We will add "terrestrial" in front of material in both cases in lines 62 and 64 to be a bit more explicit.

**Line 64: I wonder if the sentence is 100% true, as if we quantify the "export of material by rivers" it does not tell us anything about response to change. One must look at how the export varies with time.**

We will add "Changes in" to the start of the sentence to more explicitly state that it's the changes in export that reflect changes on the landscape and climate. Thank you.

**Lines 74-76: This reference might be misleading as it does not apply to all types of catchments. Does this only apply to Canada? Is this also true when 90% of the area is covered by forest? Too vague.**

Thanks for the comment. This paragraph is meant to parse out specific climate and landscape processes that add or subtract sediment and dissolved organic matter to rivers, of which this reference is a part. For example:

Wetting climate: expected to add sediment to rivers, particularly in impacted landscapes (Whitehead reference)

Drying climate: reduces runoff and erosion potential (Mosley 2015)

Impacted landscapes: ag and urban expected to contribute sediment and organic material to rivers. (St. Hilaire reference; though specific to Canada, these are universal processes).

Intact landscapes: these landscapes store nutrients, sediments, organic matter, like forested areas.

We are happy to add an additional review paper reference here to accentuate the global nature of this problem. Giri and Qiu (2016) provide a robust, international review of agricultural and urban impacts on water quality and will be added to the St. Hilaire reference.

**Line 78-80: pristine instead of intact? The sentence does not seem grammatically correct: runoff can modify surface water quality by storing water?**

We do try to avoid pristine as we would argue there's very few, if any, landscapes not impacted by humans. We feel "intact" is a better descriptor as it suggests the landscape has not been widely impacted by human interventions but still could be impacted by long-distance pollution. For the second comment, thanks, and good catch. We will re-write this sentence to, "Similarly, runoff through intact landscapes and wetlands can result in efficient soil and biological water storage and promotion of physical and biogeochemical removal of nutrients, DOC, and contaminants (Knox et al., 2008; Neary et al., 2009)."

**Line 87-89: Not clear to me how a "river network monitoring" allows distinguishing whether sediments come from tributaries or from in-river sources (banks, channels).**

Thanks for the comment, and we believe you're referring to lines 92-95, at least in the copy we submitted to the journal. Regardless, we will add the word "nested" in front of "river network" to imply monitoring of integrated river networks are best to fully understand sediment/dissolved organic matter mobilization across large river systems.

**Line 132-134: where does this data come from?**
Thanks, we will add a reference at the end of this sentence referring to the Alberta Climate Information Centre.

**Line 138-139: how big are these reservoirs? Difficult to figure out their impact. It would be nice if they appeared in Fig 1.**
These reservoirs are not particularly large on the global scale. For example, the largest 50 reservoirs in the world have a nominal volume of ~10 – 180 $km^3$. Abraham Lake (upstream NSR) has a volume of approximately 1$km^3$ while the Brazeau reservoir has a volume of approximately 0.4 $km^3$. We can only assess the impact of the dam on the NSR as we have upstream and downstream data there, compared to the Brazeau River. However, based on our data, the dam creating Abraham Lake traps substantial amounts of sediment, as is expected behind the Brazeau Dam. For dissolved organic carbon, there is very little DOC emanating from the Rocky Mountains, so Abraham Lake does not impact DOC loads any more than in other reaches of the river. DOC concentrations are higher in the Brazeau River, but it is unknown the role of the reservoir on DOC (i.e., whether DOC in water changes little and passes through the dam, or if it is photodegraded during its time in the reservoir). The reservoirs are in-fact labelled and shown in Figure 1 (Brazeau Reservoir, Abraham Lake), but we see how these could be missed. We will enhance the visibility of the labels and mention them specifically in the caption.

**Figure 1. I can't see the greyed catchment names. It would also nice to identify the 13 sites that are included in the analysis (2.3.3).**
Thank you for this observation.  We will enhance the text of the greyed catchments. The normal, dark text in that inset figure are the 13 catchments included in the analysis. We will update the caption to note this.

**Line 144: I do not recall Nippgen et al., 2011 use the catchment structural unit approach. I might be wrong.**
We included this as a foundational paper shaping the CSU approach, but you're right, this is not directly related to the CSUs used in this approach, so we have removed the reference.

**Lines 161-162: What's the origin of these data?**
The origin of the historic data are described in the preceding sentence that all flow data were collected from Water Survey of Canada and Government of Alberta. We will modify the sentence slightly to combine descriptions of the historic and more recent flow data, so that it better connects with the preceding sentence.

**Line 164: There are only 17 sites lister for water quality in Table 1, but more than 20 points in the map.**
In Figure 1, there are 17 water quality stations plotted, along with an additional four flow stations on the mainsteam of the NSR.  All tributary stations have paired quality and quantity monitoring and are shown as a single dot. For NSR mainstem stations, water quality and flow monitoring are not necessarily at the same locations, hence the separation of symbols.

**Line 165-166. I guess a rating curve I needed. A bit more info would be appreciated here.**

Thanks for the comment. We will include a reference to development of rating curves where the flow gauging is described, as follows, "Tributary monitoring occurs at the mouths of 20 WaterSHED catchments (Table 1; Figure 1) and includes: 1. Flow gauging that monitors water level and estimates river flow every 15 minutes for development of rating curves; and 2. Flow-targeted water quality…".

**Line 166. How are the predefined flow levels defined for the flow-targeted water quality sampling defined? These should be better explained.**

Thanks, this is helpful. We will modify the sentence to, "…and 2. flow-targeted water quality grab sampling (i.e., sub-monthly monitoring during spring freshet; monthly thereafter) at 30 cm depth throughout the open water season (March–October) at all sites…".

**Lines 203-205. I would appreciate more details here. I would be unable to reproduce this. Has any king of delay has been taken into consideration when flows not expected to peak at the same time?**

Thank you, we will provide more information. On a daily scale for a larger and flow-regulated river, the stations are close enough in space, and water travel is quick enough in time (i.e, typically subhourly) that changes in flow from the time of sampling would be minimal. We will add further details in the Supporting Information on how flow pairing was handled for the NSR stations not paired. Basically, we determined the catchment areas of the water quality station and the closest flow station and determined by subtraction the unmonitored area between them. We then took daily water yields (volume per area) from the closest monitored tributary and applied these to the unmonitored area to estimate flow volumes. This was then added to or subtracted from the flow station measurements depending on if the water quality station as upstream or downstream of the flow station. This would then allow us to construct a flow record at the unpaired water quality station.

**Line 215: this report is difficult to access.**

Thanks. We have updated the reference to include a URL.

**Line 215: At this point we have not seen DOC values, but as the paper only deals with 2 parameters (DOC and SSC) it would be interesting to provide more information about the data quality. I think this is necessary to interpret the results. I looked at the Laceby et al 2022 report, and it seems that 13% of the field blanc samples present DOC (environmental contamination). I would appreciate to know a bit more about this and the contamination sources.**

Thanks for your thoughtful observations here. Our river monitoring program, for which the QA/QC program is based, collects samples from more than 100 rivers province-wide, that are sampled 8-12 times per year, totalling nearly 2,000 samples over a five-year period. During that time, we collected nearly 350 field blanks from various trips across the province, including from trips that produced data for this assessment. From all trip field blank data, yes, we found contamination frequency (i.e., field blanks with hits) of 4% for TSS and 13% for DOC. Contamination frequency is an important measure, but it also matters what concentrations these hits amount to relative to environmental data. If field blank hits result in blank concentrations much lower than environmental data, one would expect the environmental data to be little impacted by potential contamination.

In our study, our rivers are relatively high in TSS (n=807; mean: 67.2 mg/L) versus the "typical" values of field blanks (B95-90 from Laceby 2022; mean: 1.3 mg/L). That amounts to ~2% potential contamination and a potential reduction down of mean concentrations from 67.2 mg/L to 66.2 mg/L. This error is very small compared to the substantial variability of TSS concentrations within a given year in a river, between years at a given river, and between rivers of a given landscape category (Figure 3). Combine that with the large variability in flow between years and rivers (i.e., influence yield/export), we further expect this field blank contamination to be inconsequential.

For DOC, our rivers are only moderately high in concentration (n=806; mean 8.9 mg/L) versus the "typical" values of field blanks (B95-90 from Laceby 2022; mean: 0.87 mg/L). This amounts to ~10% potential contamination added to samples and a potential reduction down of mean concentrations from 8.9 to 8.0 mg/L. Though we observed less variability between years and rivers for DOC compared to TSS, coefficients of variation for DOC concentrations collected from all stations was 26-94% (mean: 58%) compared to flow, which reported 32-256% coefficient of variability (mean: 143%). Therefore, we expect most of the variability in our loading and yield estimates (i.e., products of concentration and flow) to be driven by the changes in flow between years and between rivers of a given landscape category, rather than changes in DOC concentration. This gives us additional confidence that a potential contamination correction of 13% on a typical sample will not be important relative to flow changes and have even less impact on export and yield estimates. We will add a short SI section discussing this issue.

**Line 224: basins?**
Thank you. This should read, "To classify different landscape types across the NSR basin, …".

**Line 322. Strange to add references in the Results sections – where authors should only report their findings. Maybe this should be moved to the discussion section.**
Thank you. We did not see a home for this reference in the Discussion, so have decided to remove it from this sentence. This was more of an informational/interest insert into this section.

**Line 359: how is the standard error of the median computed, bootstrapping? Would it be more appropriate to report the standard error of the mean (if values normally distributed)? It is normally easier to calculate (st dev/sqrt(n)).**
Thank you for this comment. This was an error; means, as you suggest, should have been reported along with standard errors, rather than the medians. This will be changed.

**Figure 3. Are dots outliers? Information missing in the caption.**
Thank you. Yes, these are outliers and we will add the boxplot interpretation to the caption.

**Figure 4. why is TSS yield data missing for 2021 in Redwater and Tomahawk?**
Many thanks for catching this issue! These low-yield years were off the axis, likely due to erroneous axis formatting. We will fix this.

**Table S5. Correct uppercases in the table headers.**
Yes, thank you, good eye. These will be fixed.

**Figure 5. Mention in the caption which data is reported (i.e. the model used) to compute TSSs.**

Thank you for the comment. These are median loads from all four loading models. We will include this in the caption.

**Line 545. Is "proper" a common term?**
We will be more explicit and replace with "mainstem".